# Refining Deep Generative Models via Discriminator Gradient Flow

**Abdul Fatir Ansari, Ming Liang Ang & Harold Soh**
Department of Computer Science, School of Computing
National University of Singapore
{abdulfatir, angmingliang}@u.nus.edu, harold@comp.nus.edu.sg

## Abstract

Deep generative modeling has seen impressive advances in recent years, to the point where it is now commonplace to see simulated samples (e.g., images) that closely resemble real-world data. However, generation quality is generally inconsistent for any given model and can vary dramatically between samples. We introduce Discriminator Gradient $f$low (DG$f$low), a new technique that improves generated samples via the gradient flow of entropy-regularized $f$-divergences between the real and the generated data distributions. The gradient flow takes the form of a *non-linear* Fokker-Plank equation, which can be easily simulated by sampling from the equivalent McKean-Vlasov process. By refining inferior samples, our technique avoids wasteful sample rejection used by previous methods (DRS & MH-GAN). Compared to existing works that focus on specific GAN variants, we show our refinement approach can be applied to GANs with vector-valued critics and even other deep generative models such as VAEs and Normalizing Flows. Empirical results on multiple synthetic, image, and text datasets demonstrate that DG$f$low leads to significant improvement in the quality of generated samples for a variety of generative models, outperforming the state-of-the-art Discriminator Optimal Transport (DOT) and Discriminator Driven Latent Sampling (DDLS) methods.

## 1 Introduction

Deep generative models (DGMs) have excelled at numerous tasks, from generating realistic images (Brock et al., 2019) to learning policies in reinforcement learning (Ho & Ermon, 2016). Among the variety of proposed DGMs, Generative Adversarial Networks (GANs) (Goodfellow et al., 2014) have received widespread popularity for their ability to generate high quality samples that resemble real data. Unlike Variational Autoencoders (VAEs) (Kingma & Welling, 2014) and Normalizing Flows (Rezende & Mohamed, 2015; Kingma & Dhariwal, 2018), GANs are likelihood-free methods; training is formulated as a minimax optimization problem involving a generator and a discriminator. The generator seeks to generate samples that are similar to the real data by minimizing a measure of discrepancy (between the generated samples and real samples) furnished by the discriminator. The discriminator is trained to distinguish the generated samples from the real samples. Once trained, the generator is used to simulate samples and the discriminator has traditionally been discarded.

However, recent work has shown that discarding the discriminator is wasteful — it actually contains useful information about the underlying data distribution. This insight has led to *sample improvement* techniques that use this information to improve the quality of generated samples (Azadi et al., 2019; Turner et al., 2019; Tanaka, 2019; Che et al., 2020). Unfortunately, current methods either rely on wasteful rejection operations in the data space (Azadi et al., 2019; Turner et al., 2019), or require a sensitive diffusion term to ensure sample diversity (Che et al., 2020). Prior work has also focused on improving GANs with scalar-valued discriminators, which excludes a large family of GANs with vector-valued critics, e.g., MMDGAN (Li et al., 2017; Bińkowski et al., 2018) and OCFGAN (Ansari et al., 2020), and likelihood-based generative models.

In this work, we propose Discriminator Gradient $f$low (DG$f$low) which formulates sample improvement as refining inferior samples using the *gradient flow* of $f$-divergences between the generator and the real data distributions (Fig. 1). DG$f$low avoids wasteful rejection operations and can be used in a deterministic setting without a diffusion term. Existing state-of-the-art methods — specifically, Discriminator Optimal Transport (DOT) (Tanaka, 2019) and Discriminator Driven Latent Sampling (DDLS) (Che et al., 2020) — can be viewed as special cases of DG$f$low. Similar to DDLS, DG$f$low recovers the real data distribution when the gradient flow is simulated exactly.

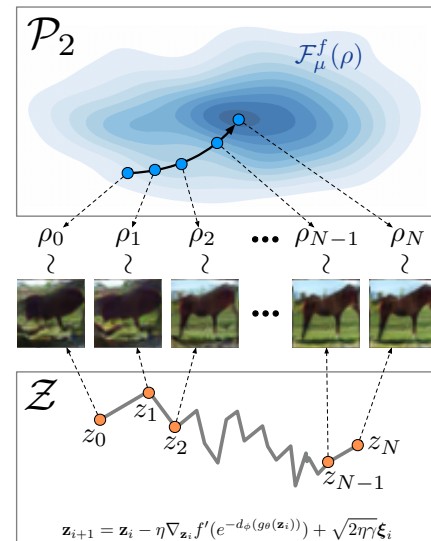

Figure 1: An illustration of refinement using DG$f$low, with the gradient flow in the 2-Wasserstein space $\mathcal{P}_2$ (top) and the corresponding discretized SDE in the latent space $\mathcal{Z}$ (bottom). The image samples from the densities along the gradient flow are shown in the middle.

We further present a generalized framework that employs existing pre-trained discriminators to refine samples from a *variety* of deep generative models: we demonstrate our method can be applied to GANs with vector-valued critics, and even likelihood-based models such as VAEs and Normalizing Flows. Empirical results on synthetic datasets, and benchmark image (CIFAR10, STL10) and text (Billion Words) datasets demonstrate that our gradient flow-based approach outperforms DOT and DDLS on multiple quantitative evaluation metrics.

In summary, this paper's key contributions are:

- DG$f$low, a method to refine deep generative models using the gradient flow of $f$-divergences;

- a framework that extends DG$f$low to GANs with vector-valued critics, VAEs, and Normalizing Flows;

- experiments on a variety of generative models trained on synthetic, image (CIFAR10 & STL10), and text (Billion Words) datasets demonstrating that DG$f$low is effective in improving samples from generative models.

## 2 BACKGROUND: GRADIENT FLOWS

The following gives a brief introduction to gradient flows; we refer readers to the excellent overview by Santambrogio (2017) for a more thorough introduction.

Let $(\mathcal{X}, \|\cdot\|_2)$ be a Euclidean space and $F : \mathcal{X} \to \mathbb{R}$ be a smooth energy function. The gradient flow of $F$ is the smooth curve $\{\mathbf{x}_t\}_{t \in \mathbb{R}_+}$ that follows the direction of steepest descent, i.e.,

$$\mathbf{x}'(t) = -\nabla F(\mathbf{x}(t)). \tag{1}$$

The value of the energy $F$ is minimized along this curve. This idea of steepest descent curves can be characterized in arbitrary metric spaces via the *minimizing movement scheme* (Jordan et al., 1998). Of particular interest is the metric space of probability measures that is endowed with the Wasserstein distance ($\mathcal{W}_p$); the Wasserstein distance is a metric and the $\mathcal{W}_p$ topology satisfies weak convergence of probability measures (Villani, 2008, Theorem 6.9). Gradient flows in the 2-Wasserstein space $(\mathcal{P}_2(\Omega), \mathcal{W}_2)$ — i.e., the space of probability measures with finite second moments and the 2-Wasserstein metric — have been studied extensively. Let $\{\rho_t\}_{t \in \mathbb{R}_+}$ be the gradient flow of a functional $\mathcal{F}$ in the 2-Wasserstein space, where $\rho_t$ is absolutely continuous with respect to the Lebesgue measure. The curve $\{\rho_t\}_{t \in \mathbb{R}_+}$ satisfies the continuity equation (Ambrosio et al., 2008, Theorem 8.3.1),

$$\partial_t \rho_t + \nabla \cdot (\rho_t \mathbf{v}_t) = 0. \tag{2}$$

The velocity field $\mathbf{v}_t$ in Eq. (2) is given by

$$\mathbf{v}_t(\mathbf{x}) = -\nabla_{\mathbf{x}} \frac{\delta \mathcal{F}}{\delta \rho}(\rho), \tag{3}$$

where $\frac{\delta \mathcal{F}}{\delta \rho}$ denotes the first variation of the functional $\mathcal{F}$.

Since the seminal work of Jordan et al. (1998) that showed that the Fokker-Plank equation is the gradient flow of a particular functional in the Wasserstein space, gradient flows in the Wasserstein metric have been a popular tool in the analysis of partial differential equations (PDEs). For example, they have been applied to the study of the porous-medium equation (Otto, 2001), crowd modeling (Maury et al., 2010; 2011), and mean-field games (Almulla et al., 2017). More recently, gradient flows of various distances used in deep generative modeling literature have been proposed, notably that of the sliced Wasserstein distance (Liutkus et al., 2019), the maximum mean discrepancy (Arbel et al., 2019), the Stein discrepancy (Liu, 2017), and the Sobolev discrepancy (Mroueh et al., 2019). Gradient flows have also been used for learning non-parametric and parametric implicit generative models (Liutkus et al., 2019; Gao et al., 2019; 2020). As an example of the latter, Variational Gradient Flow (Gao et al., 2019) learns a mapping between latent vectors and samples evolved using the gradient flow of $f$-divergences. In this work, we present a method using gradient flows of entropy-regularized $f$-divergences for refining samples from deep generative models employing existing discriminators as density-ratio estimators.

## 3 GENERATOR REFINEMENT VIA DISCRIMINATOR GRADIENT FLOW

This section describes our main contribution: Discriminator Gradient $f$low (DG$f$low). As an overview, we begin with the construction of the gradient flow of entropy-regularized $f$-divergences and describe its application to sample refinement. We then discuss how to simulate the gradient flow in the latent space of the generator — a procedure more suitable for high-dimensional datasets. Finally, we present a simple technique that extends our method to generative models that have not yet been studied in the context of refinement. Due to space constraints, we focus on conveying the key concepts and relegate details (e.g., proofs) to the appendix.

The entropy-regularized $f$-divergence functional is defined as

$$\mathcal{F}_\mu^f(\rho) \triangleq \mathcal{D}_f(\mu\|\rho) - \gamma\mathcal{H}(\rho), \tag{4}$$

where the $f$-divergence term $\mathcal{D}_f(\mu\|\rho)$ ensures that the "distance" between the probability density $\rho$ and the target density $\mu$ decreases along the gradient flow. The differential entropy term $\mathcal{H}(\rho)$ improves diversity and expressiveness when the gradient flow is simulated for finite time-steps. We now construct the gradient flow of $\mathcal{F}_\mu^f$.

**Lemma 3.1.** *Define the functional $\mathcal{F}_\mu^f : \mathcal{P}_2(\Omega) \to \mathbb{R}$ as*

$$\mathcal{F}_\mu^f(\rho) \triangleq \underbrace{\int f\left(\rho(\mathbf{x})/\mu(\mathbf{x})\right)\mu(\mathbf{x})d\mathbf{x}}_{\text{f-divergence}} + \gamma\underbrace{\int \rho(\mathbf{x})\log\rho(\mathbf{x})d\mathbf{x}}_{\text{negative entropy}}, \tag{5}$$

*where $f$ is a twice-differentiable convex function with $f(1) = 0$. The gradient flow of the functional $\mathcal{F}_\mu^f(\rho)$ in the Wasserstein space $(P_2(\Omega), \mathcal{W}_2)$ is given by the following PDE,*

$$\partial_t\rho_t(\mathbf{x}) - \nabla_{\mathbf{x}} \cdot (\rho_t(\mathbf{x})\nabla_{\mathbf{x}}f'\left(\rho_t(\mathbf{x})/\mu(\mathbf{x})\right)) - \gamma\Delta_{\mathbf{xx}}\rho_t(\mathbf{x}) = 0, \tag{6}$$

*where $\nabla_{\mathbf{x}}\cdot$ and $\Delta_{\mathbf{xx}}$ denote the divergence and the Laplace operators respectively.*

The proof is given in Appendix A.1. The PDE in Eq. (6) is a type of Fokker-Plank equation (FPE). FPEs have been studied extensively in the literature of stochastic processes and have a Stochastic Differential Equation (SDE) counterpart (Risken, 1996). In the case of Eq. (6), the equivalent SDE is given by

$$d\mathbf{x}_t = \underbrace{-\nabla_{\mathbf{x}}f'\left(\rho_t/\mu\right)(\mathbf{x}_t)dt}_{\text{drift}} + \underbrace{\sqrt{2\gamma}d\mathbf{w}_t}_{\text{diffusion}}, \tag{7}$$

where $d\mathbf{w}_t$ denotes the standard Wiener process. Eq. (7) defines the evolution of a particle $\mathbf{x}_t$ under the influence of drift and diffusion. Specifically, it is a McKean-Vlasov process (Braun & Hepp, 1977) which is a type of *non-linear* stochastic process as the drift term at any time $t$ depends on the distribution $\rho_t$ of the particle $\mathbf{x}_t$. Eqs. (6) and (7) are equivalent in the sense that the distribution of the particle $\mathbf{x}_t$ in Eq. (7) solves the PDE in Eq. (6). Consequently, samples from the density $\rho_t$ along the gradient flow can be obtained by first drawing samples $\mathbf{x}_0 \sim \rho_0$ and then simulating the

SDE in Eq. (7). The SDE can be approximately simulated via the stochastic Euler scheme (also known as the Euler-Maruyama method) (Beyn & Kruse, 2011) given by

$$\mathbf{x}_{\tau_{n+1}} = \mathbf{x}_{\tau_n} - \eta \nabla_{\mathbf{x}} f'\left(\rho_{\tau_n}/\mu\right)\left(\mathbf{x}_{\tau_n}\right) + \sqrt{2\gamma\eta}\boldsymbol{\xi}_{\tau_n}, \tag{8}$$

where $\boldsymbol{\xi}_{\tau_n} \sim \mathcal{N}(\mathbf{0}, \mathbf{I})$, the time interval $[0, T]$ is partitioned into equal intervals of size $\eta$ and $\tau_0 < \tau_1 < \cdots < \tau_N$ denote the discretized time-steps.

Eq. (8) provides a non-parametric procedure to refine samples from a generator $g_\theta$ where we let $\mu$ be the density of real samples and $\rho_{\tau_0}$ the density of samples generated from $g_\theta$ obtained by first sampling from the prior latent distribution $\mathbf{z} \sim p_Z(\mathbf{z})$ and then feeding $\mathbf{z}$ into $g_\theta$. We first generate particles $\mathbf{x}_0 \sim \rho_{\tau_0}$ and then update the particles using Eq. (8) for $N$ time steps.

Given a binary classifier (discriminator) $D$ that has been trained to distinguish between samples from $\mu$ and $\rho_{\tau_0}$, the density-ratio $\rho_{\tau_0}(\mathbf{x})/\mu(\mathbf{x})$ can be estimated via the well-known *density-ratio trick* (Sugiyama et al., 2012),

$$\rho_{\tau_0}(\mathbf{x})/\mu(\mathbf{x}) = \frac{1 - D(y = 1|\mathbf{x})}{D(y = 1|\mathbf{x})} = \exp(-d(\mathbf{x})), \tag{9}$$

where $D(y = 1|\mathbf{x})$ denotes the conditional probability of the sample $\mathbf{x}$ being from $\mu$ and $d(\mathbf{x})$ denotes the logit output of the classifier $D$. We term this procedure where samples are refined via gradient flow of $f$-divergences as Discriminator Gradient $f$low (DG$f$low).

### 3.1 REFINEMENT IN THE LATENT SPACE

Eq. (8) requires a running estimate of the density-ratio $\rho_{\tau_n}(\mathbf{x})/\mu(\mathbf{x})$, which can be approximated using the *stale estimate* $\rho_{\tau_n}(\mathbf{x})/\mu(\mathbf{x}) \approx \rho_{\tau_0}(\mathbf{x})/\mu(\mathbf{x})$ for $\eta \to 0$ and small $N$, where the density $\rho_{\tau_n}$ will be close to $\rho_{\tau_0}$. However, our initial image experiments showed that refining directly in high-dimensional data-spaces with the stale estimate is problematic; error is accumulated at each time-step leading to a visible degradation in the quality of data samples (e.g., appearance of artifacts in images).

To tackle this problem, we propose refining the latent vectors before mapping them to samples in data-space using $g_\theta$. We describe a procedure analogous to Eq. (8) but in the latent space for generators $g_\theta$ that take a latent vector $\mathbf{z} \in \mathcal{Z}$ as input and generate a sample $\mathbf{x} \in \mathcal{X}$. We first show in Lemma 3.2 that the density-ratio in the latent space between two distributions can be estimated via the density-ratio of corresponding distributions in the data space.

**Lemma 3.2.** *Let $g : \mathcal{Z} \to \mathcal{X}$ be a sufficiently well-behaved injective function where $\mathcal{Z} \subseteq \mathbb{R}^n$ and $\mathcal{X} \subset \mathbb{R}^m$ with $m > n$. Let $p_Z(\mathbf{z})$, $p_{\hat{Z}}(\hat{\mathbf{z}})$ be probability densities on $\mathcal{Z}$ and $q_X(\mathbf{x})$, $q_{\hat{X}}(\hat{\mathbf{x}})$ be the densities of the pushforward measures $g\sharp Z$, $g\sharp\hat{Z}$ respectively. Assume that $p_Z(\mathbf{z})$ and $p_{\hat{Z}}(\hat{\mathbf{z}})$ have same support, and the Jacobian matrix $\mathbf{J}_g$ has full column rank. Then, the density-ratio $p_{\hat{Z}}(\mathbf{u})/p_Z(\mathbf{u})$ at the point $\mathbf{u} \in \mathcal{Z}$ is given by*

$$\frac{p_{\hat{Z}}(\mathbf{u})}{p_Z(\mathbf{u})} = \frac{q_{\hat{X}}(g(\mathbf{u}))}{q_X(g(\mathbf{u}))}. \tag{10}$$

The proof is in Appendix A.2. We let $p_{\hat{Z}}(\hat{\mathbf{z}})$ be the density of the "correct" latent space distribution induced by a generator $g_\theta$, i.e., $p_{\hat{Z}}(\hat{\mathbf{z}})$ is the density of a probability measure whose pushforward under $g_\theta$ approximately equals the target data density $\mu$. The density-ratio of the prior latent distribution $p_Z(\mathbf{z})$ and $p_{\hat{Z}}(\hat{\mathbf{z}})$ can now be computed by combining Lemma 3.2 with Eq. (9),

$$\frac{p_Z(\mathbf{u})}{p_{\hat{Z}}(\mathbf{u})} = \frac{\rho_{\tau_0}(g_\theta(\mathbf{u}))}{\mu(g_\theta(\mathbf{u}))} = \exp(-d(g_\theta(\mathbf{u}))). \tag{11}$$

---

**Algorithm 1** Refinement in the Latent Space using DG$f$low.

---

**Require:** First derivative of $f$ ($f'$), generator ($g_\theta$), discriminator ($d_\phi$), number of update steps ($N$), step-size ($\eta$), noise factor ($\gamma$).

1: $\mathbf{z}_0 \sim p_Z(\mathbf{z})$         $\triangleright$ Sample from the prior.
2: **for** $i \leftarrow 0, N$ **do**
3:      $\boldsymbol{\xi}_i \sim \mathcal{N}(0, I)$
4:      $\mathbf{z}_{i+1} = \mathbf{z}_i - \eta \nabla_{\mathbf{z}_i} f'(e^{-d_\phi(g_\theta(\mathbf{z}_i))}) + \sqrt{2\eta\gamma}\boldsymbol{\xi}_i$
5: **end for**
6: **return** $g_\theta(\mathbf{z}_n)$      $\triangleright$ The refined sample.

---

Although a generator $g_\theta$ parameterized by a neural network may not satisfy the conditions of injectivity and full column rank Jacobian matrix $\mathbf{J}_{g_\theta}$, Eq. (11) provides an approximation that works well in practice as shown by our experiments. Combining Eq. (11) with Eq. (8) provides us with an update rule for refining samples in the latent space,

$$\mathbf{u}_{\tau_{n+1}} = \mathbf{u}_{\tau_n} - \eta \nabla_{\mathbf{u}} f' \left( p_{\mathbf{u}_{\tau_n}} / p_{\hat{Z}} \right) (\mathbf{u}_{\tau_n}) + \sqrt{2\gamma\eta} \boldsymbol{\xi}_{\tau_n}, \tag{12}$$

where $\mathbf{u}_{\tau_0} \sim p_Z(\mathbf{z})$ and the density-ratio $p_{\mathbf{u}_{\tau_n}} / p_{\hat{Z}}$ is approximated using the stale estimate $p_{\mathbf{u}_{\tau_0}} / p_{\hat{Z}} = \exp(-d(g_\theta(\mathbf{u})))$. We summarize the complete algorithm in Algorithm 1.

## 3.2 REFINEMENT FOR ALL

Thus far, prior work (Azadi et al., 2019; Turner et al., 2019; Tanaka, 2019; Che et al., 2020) has focused on improving samples for GANs with scalar-valued discriminators, which comprises the canonical GAN as well as recent variants, e.g., WGAN (Gulrajani et al., 2017), and SNGAN (Miyato et al., 2018). Here, we propose a technique that extends our approach to refine samples from a larger class of DGMs including GANs with vector-valued critics, VAEs, and Normalizing Flows.

Let $p_\theta$ be the density of the samples generated by a generator $g_\theta$ and $\mu$ be the density of the real data distribution. We are interested in refining samples from $g_\theta$; however, a corresponding density-ratio estimator for $p_\theta / \mu$ is unavailable, as is the case with the aforementioned generative models.

Let $D_\phi$ be a discriminator that has been trained on the same dataset but for a different generative model $g_\phi$ (e.g., let $g_\phi$ and $D_\phi$ be the generator and discriminator of SNGAN respectively). $D_\phi$ can be used to compute the density ratio $p_\phi / \mu$. A straightforward technique would be to use the crude approximation $p_\theta / \mu \approx p_\phi / \mu$, which could work provided $p_\theta$ and $p_\phi$ are not too far from each other. Our experiments show that this simple approximation works to a limited extent (see appendix E).

To improve upon the crude approximation above, we propose to correct the density-ratio estimate. Specifically, a discriminator $D_\lambda$ is initialized with the weights from $D_\phi$ and is fine-tuned on samples from $g_\phi$ and $g_\theta$. $D_\phi$ and $D_\lambda$ are then used to approximate the density-ratio $p_\theta / \mu$,

$$\frac{p_\theta(\mathbf{x})}{\mu(\mathbf{x})} = \frac{p_\phi(\mathbf{x})}{\mu(\mathbf{x})} \frac{p_\theta(\mathbf{x})}{p_\phi(\mathbf{x})} = \exp(-d_\phi(\mathbf{x})) \cdot \exp(-d_\lambda(\mathbf{x})), \tag{13}$$

where $d_\phi$ and $d_\lambda$ are logits output from $D_\phi$ and $D_\lambda$, respectively. We term the network $D_\lambda$ the *density ratio corrector*, which experiments show produces higher quality samples than using $p_\theta / \mu \approx p_\phi / \mu$. The estimate in Eq. (13) is similar to telescoping density-ratio estimation (TRE), a technique proposed in very recent independent work (Rhodes et al., 2020). In brief, Rhodes et al. (2020) show that classifier-based density ratio estimators perform poorly when distributions are "too far apart"; the classifier can easily distinguish between the distributions, even with a poor estimate of the density ratio. TRE expands the standard density ratio into a telescoping product of more difficult-to-distinguish intermediate density ratios. Likewise, in Eq. (13), we treat $p_\phi$ as an intermediate distribution and estimate the final density-ratio as a product of two density-ratios.

## 4 RELATED WORK

Azadi et al. (2019) first proposed the idea of improving samples from a GAN's generator by discriminator rejection sampling (DRS), making use of the density-ratio provided by the discriminator to estimate the acceptance probability. Metropolis-Hastings GAN (MH-GAN) (Turner et al., 2019) improved upon the costly rejection sampling procedure via the Metropolis-Hastings algorithm. Unlike DG$f$low, both of these methods reject inferior samples instead of refining them.

Our method is closely related to recent state-of-the-art sample refinement techniques, specifically Discriminator-Driven Latent Sampling (DDLS) (Che et al., 2020) and Discriminator Optimal Transport (DOT) (Tanaka, 2019). In fact, both these methods can be seen as special cases of DG$f$low.

DDLS treats a GAN as an energy-based model and uses Langevin dynamics to sample from the energy-based latent distribution $p_t(\mathbf{z}) \propto p_Z(\mathbf{z}) \exp(d(g_\theta(\mathbf{z})))$ induced by performing rejection sampling in the latent space. This distribution is the same as $p_{\hat{Z}}(\hat{\mathbf{z}})$, which can be seen by rearranging terms in Eq. (11). If we use the KL-divergence by setting $f = r \log r$, DG$f$low is equivalent to DDLS. However, there are practical differences that make DG$f$low more appealing. DDLS requires estimation of the score function $\nabla_{\mathbf{z}} \{ \log p_Z(\mathbf{z}) + d(g_\theta(\mathbf{z})) \}$ to perform the update which becomes

undefined if $\mathbf{z}$ escapes the support of $p_Z(\mathbf{z})$, e.g., in the case of the uniform prior distribution commonly used in GANs; handling such cases would require techniques such as projected gradient descent. This problem does not arise in the case of DG$f$low since it only uses the density-ratio that is implicitly defined by the discriminator. Moreover, DDLS uses Langevin dynamics which *requires* the sensitive diffusion term to ensure diversity and to prevent points from collapsing to the maximum-likelihood point. In DG$f$low, the sample diversity is ensured by the density-ratio term and the diffusion term serves as an enhancement. Note that DG$f$low performs well even without the diffusion term (i.e., with $\gamma = 0$, see Tables 13 & 14 in the appendix). This deterministic variant of DG$f$low is a practical alternative with one less hyperparameter to tune.

DOT refines samples by constructing an Optimal Transport (OT) map induced by the WGAN discriminator. The OT map is realized by means of a deterministic optimization problem in the vicinity of the generated samples. If we further analyze the case of DG$f$low with $\gamma = 0$ and solve the resulting ordinary differential equation (ODE) using the backward Euler method,

$$\mathbf{u}_{\tau_{n+1}} = \operatorname*{arg\,min}_{\mathbf{u} \in \mathbb{R}^n} \left\{ f'\left(p_{\mathbf{u}_{\tau_n}}/p_{\hat{Z}}\right)(\mathbf{u}) + \frac{1}{2\lambda}\|\mathbf{u} - \mathbf{u}_{\tau_n}\|^2 \right\}, \tag{14}$$

DOT emerges as a special case when we consider a single update step of Eq. (14) using gradient descent and set $f'(t) = \log(t)^1$ with $\lambda = \frac{1}{2}$. This connection of DG$f$low to DOT, an optimal transport technique, is perhaps unsurprising given the relationship between gradient flows and the dynamical Benamou-Brenier formulation of optimal transport (Santambrogio, 2017).

Recent work has also sought to improve generative models via sample evolution in the training/generation process. In energy-based generative models (Arbel et al., 2021; Deng et al., 2020), the energy functions can be viewed as a component that improves some base generator. For example, the Generalized Energy-Based Model (GEBM) (Arbel et al., 2021) jointly trains a base generator by minimizing a lower bound of the KL divergence along with an energy function in an alternating fashion. Once trained, the energy function is used to refine samples from the base generator using Langevin dynamics and serves a similar purpose to the discriminator in DDLS and DG$f$low. The Noise Conditional Score Network (NCSN) (Song & Ermon, 2019; 2020) — a score-based generative model — can be seen as a gradient flow that refines a sample right from noise to data. Latent Optimization GAN (LOGAN) (Wu et al., 2020) optimizes a latent vector via natural gradient descent as part of the GAN training process. In contrast to these works, we primarily focus on refining samples from *pretrained* generative models using the gradient flow of $f$-divergences.[2]

## 5 EXPERIMENTS

In this section, we present empirical results on various deep generative models trained on multiple synthetic and real world datasets. Our primary goals were to determine if (a) DG$f$low is effective in improving the quality of samples from generative models, (b) the proposed extension to other generative models improves their sample quality, and (c) DG$f$low is generalizable to different types of data and metrics. Note that we did not seek to achieve state-of-the-art results for the datasets studied but to demonstrate that DG$f$low is able to significantly improve samples from the bare generators for different models.

We experimented with three $f$-divergences, namely the Kullback-Leibler (KL) divergence, the Jensen-Shannon (JS) divergence, and the $\log$ D divergence (Gao et al., 2019). The specific forms of the functions $f$ and corresponding derivatives are tabulated in Table 7 (appendix). We compare DG$f$low with two state-of-the-art competing methods: DOT and DDLS. In this section we discuss the main results and relegate details to the appendix. Our code is available online at `https://github.com/clear-nus/DGflow`.

### 5.1 2D DATASETS

We first tested DG$f$low on two synthetic datasets, 25Gaussians and 2DSwissroll, to visually inspect the improvement in the quality of generated samples. We generated 5000 samples from a trained WGAN-GP generator and refined them using DOT, DDLS, and DG$f$low. We performed refinement in the latent space for DDLS and directly in the data-space for DOT and DG$f$low. Fig. 2 shows

---

[1]This implies that $f(t) = t\log t - t + 1$, which is a twice-differentiable convex function with $f(0) = 1$.
[2]For further discussion about these techniques, please refer to appendix C.

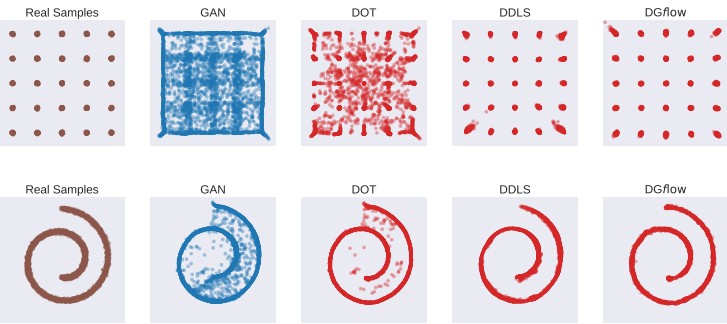

Figure 2: Qualitative comparison of DG$f$low$_{(KL)}$ with DOT and DDLS on synthetic 2D datasets.

the samples generated from the WGAN-GP generator (blue) and the refined samples using different techniques (red) against the real samples from the training dataset (brown). Although the WGAN-GP generator learned the overall structure of the dataset, it also learned a number of spurious modes. DOT is able to refine the spurious samples but to a limited degree. In contrast, DDLS and DG$f$low are able to correct almost all spurious samples and are able to recover the correct structure of the data. Visualizations for DG$f$low with different $f$-divergences can be found in the appendix (Fig. 4).

We also compared the different methods quantitatively on two metrics: % high quality samples and kernel density estimate (KDE) score. A sample is classified as a high quality sample if it lies within 4 standard deviations of its nearest Gaussian. The KDE score is computed by first estimating the KDE using generated samples and then computing the log-likelihood of the training samples under the KDE estimate. We computed both the metrics 10 times using 5000 samples and report the mean in Table 1. The quantitative metrics reinforce the qualitative analysis and show that DDLS and DG$f$low significantly improve the samples from the generator, with DG$f$low performing slightly better than DDLS in terms of the KDE score.

Table 1: Quantitative comparison on the 25Gaussians dataset. Higher scores are better.

|  | % High Quality | KDE Score |
|---|---|---|
| GAN | 26.5 ± .8 | -7037 ± 64 |
| DOT | 69.8 ± .7 | -4149 ± 39 |
| DDLS | **89.3 ± .6** | -2997 ± 17 |
| DG$f$low$_{(KL)}$ | **89.5 ± .4** | **-2893 ± 07** |
| DG$f$low$_{(JS)}$ | 82.6 ± .4 | -3118 ± 19 |
| DG$f$low$_{(\log D)}$ | 84.5 ± .3 | -3036 ± 14 |

## 5.2 IMAGE EXPERIMENTS

We conducted experiments on the CIFAR10 and STL10 datasets to demonstrate the efficacy of DG$f$low in the real-world setting. We followed the setup of Tanaka (2019) for our image experiments. We used the Fréchet Inception Distance (FID) (Heusel et al., 2017) and Inception Score (IS) (Salimans et al., 2016) metrics to evaluate the quality of generated samples before and after refinement. A high value of IS and a low value of FID corresponds to high quality samples, respectively.

We first applied DG$f$low to GANs with scalar-valued discriminators (e.g., WGAN-GP, SNGAN) trained on the CIFAR10 and the STL10 datasets. Table 2 shows that DG$f$low significantly improves the quality of

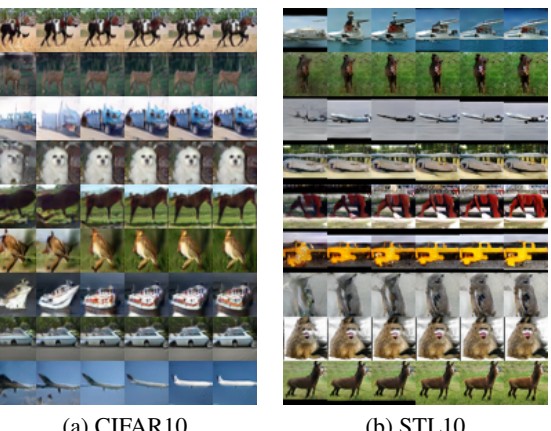

(a) CIFAR10        (b) STL10

Figure 3: Improvement in the quality of samples generated from the base model (leftmost columns) over the steps of DG$f$low for SN-ResNet-GAN and SN-DCGAN on the CIFAR10 and STL10 datasets respectively.

Table 2: Comparison of different variants of DG$f$low with DOT on the CIFAR10 and STL10 datasets. For SN-DCGAN, (hi) denotes the hinge loss and (ns) denotes the non-saturating loss. Lower scores are better. DG$f$low's results have been averaged over 5 random runs with the standard deviation in parentheses.

| | Model | Fréchet Inception Distance | | | | |
| | | Base Model | DOT | DG$f$low$_{(KL)}$ | DG$f$low$_{(JS)}$ | DG$f$low$_{(\log D)}$ |
|---|---|---|---|---|---|---|
| CIFAR10 | WGAN-GP | 28.37 (.08) | 24.14 | 24.68 (.09) | **23.15 (.07)** | 24.53 (.11) |
| | SN-DCGAN (hi) | 20.70 (.05) | 17.12 | **15.68 (.07)** | 16.45 (.06) | 17.36 (.05) |
| | SN-DCGAN (ns) | 20.90 (.11) | 15.78 | **15.30 (.08)** | 15.90 (.11) | 16.42 (.05) |
| | SN-ResNet-GAN | 14.10 (.06) | – | **9.62 (.03)** | 9.79 (.02) | 9.73 (.05) |
| STL10 | WGAN-GP | 51.50 (.15) | 44.45 | **39.07 (.07)** | 50.83 (.06) | 39.71 (.29) |
| | SN-DCGAN (hi) | 40.54 (.17) | **34.85** | 34.95 (.06) | 36.37 (.12) | 36.56 (.08) |
| | SN-DCGAN (ns) | 41.86 (.12) | 34.84 | **34.60 (.11)** | 35.37 (.12) | 37.07 (.14) |

Table 3: Inception scores of different generative models, DRS, MH-GAN, DDLS, and DG$f$low on the CIFAR10 dataset. Higher scores are better.

| Model | Inception Score |
|---|---|
| WGAN-GP (Gulrajani et al., 2017) | 7.86 (.07) |
| ProgressiveGAN (Karras et al., 2017) | 8.80 (.05) |
| SN-ResNet-GAN (Miyato et al., 2018) | 8.22 (.05) |
| NCSN (Song & Ermon, 2019) | 8.87 (.12) |
| DCGAN | 2.88 |
| DCGAN + DRS (cal) (Azadi et al., 2019) | 3.07 |
| DCGAN + MH (cal) (Turner et al., 2019) | 3.38 |
| SN-ResNet-GAN (our evaluation) | 8.38 (.03) |
| SN-ResNet-GAN + DDLS (cal) (Che et al., 2020) | 9.09 (.10) |
| SN-ResNet-GAN + DG$f$low$_{(KL)}$ | **9.35 (.03)** |
| BigGAN | 9.22 |

the samples in terms of the FID score and outperforms DOT on multiple models. The corresponding values of the Inception score can be found in the Appendix (Table 11), which shows DG$f$low outperforms DOT on all models. In Table 3, we reproduce previously reported IS results for generative models and other sample improvement methods (DRS, MH-GAN, and DDLS) for completeness. DG$f$low performs the best in terms of relative improvement from the base score and even outperforms the state-of-the-art BigGAN (Brock et al., 2019), a conditional generative model, without the need for additional labels. Qualitatively, DG$f$low improves the vibrance of the samples and corrects deformations in the foreground object. Fig. 3 shows the change in the quality of samples when using DG$f$low where the leftmost columns show the image generated form the base models and the successive columns show the refined sample using DG$f$low over increments of 5 update steps.

We then evaluated the ability of DG$f$low to refine samples from generative models *without corresponding discriminators*, namely MMDGAN, OCFGAN-GP, VAEs, and Normalizing Flows (Glow). We used the SN-DCGAN (ns) as the surrogate discriminator $D_\phi$ for these models and fine-tuned density ratio correctors $D_\lambda$ for each model as described in section 3.2. Table 4 shows the FID scores achieved by these models without and with refinement using DG$f$low. We obtain a clear improvement in quality of samples when these generative models are combined with DG$f$low.

### 5.3 CHARACTER-LEVEL LANGUAGE MODELING

Finally, we conducted an experiment on the character-level language modeling task proposed by Gulrajani et al. (2017) to show that DG$f$low works on different types of data. We trained a character-level GAN language model on the Billion Words Dataset (Chelba et al., 2013), which was pre-processed into 32-character long strings. We evaluated the generated samples using the JS-4 and JS-6 scores which compute the Jensen-Shannon divergence between the 4-gram and 6-gram probabilities of the data generated by the model and the real data. Table 5 (a) shows that DG$f$low leads to

Table 4: Comparison of different variants of DG$f$low applied to MMDGAN, OCFGAN-GP, VAE, and Glow models. Lower scores are better. Results have been averaged over 5 random runs with the standard deviation in parentheses.

| | Model | Fréchet Inception Distance | | | |
|---|---|---|---|---|---|
| | | Base Model | DG$f$low$_{(KL)}$ | DG$f$low$_{(JS)}$ | DG$f$low$_{(\log D)}$ |
| CIFAR10 | MMDGAN | 41.98 (.12) | **36.75 (.09)** | 38.06 (.14) | 37.75 (.10) |
| | OCFGAN-GP | 31.98 (.12) | **26.89 (.06)** | 28.20 (.06) | 27.82 (.09) |
| | VAE | 129.5 (.13) | **116.0 (.21)** | 128.9 (.13) | 115.2 (.06) |
| | Glow | 100.5 (.52) | **79.02 (.23)** | 94.61 (.34) | 81.12 (.35) |
| STL10 | MMDGAN | 47.20 (.07) | 43.21 (.06) | 46.74 (.05) | **43.06 (.05)** |
| | OCFGAN-GP | 36.55 (.08) | 31.12 (.13) | 36.05 (.11) | **30.61 (.14)** |
| | VAE | 150.5 (.09) | **130.1 (.18)** | 149.9 (.08) | 132.5 (.28) |

Table 5: Results of DG$f$low on a character-level GAN language model.

(a) JS-4 and JS-6 scores. Lower scores are better.

| Model | JS-4 | JS-6 |
|---|---|---|
| WGAN-GP | 0.224 (.0009) | 0.574 (.0015) |
| DG$f$low$_{(KL)}$ | 0.212 (.0008) | 0.512 (.0012) |
| DG$f$low$_{(JS)}$ | **0.186** (.0007) | 0.508 (.0011) |
| DG$f$low$_{(\log D)}$ | 0.209 (.0005) | **0.506** (.0008) |

(b) Examples of text samples refined by DG$f$low.

| Generated by WGAN-GP | Refined by DG$f$low |
|---|---|
| In Ruoduce that fhance would pol | In product that chance could rol |
| I said thowe toot lind talker . | I said this stood line talked 10 |
| Now their rarning injurer hows | Now their warning injurer shows |
| Police report in B0sbu does off | Police report inturner will befe |
| We gine jaid 121 , one bub like | We gave wall said left out like |
| In years in 19mbisuch said he h | In years in 1900b such said he h |

an improvement in the JS-4 and JS-6 scores. Table 5 (b) shows example sentences where DG$f$low visibly improves the quality of generated text.

## 6 CONCLUSION

In this paper, we proposed a technique to improve samples from deep generative models by refining them using gradient flow of $f$-divergences between the real and the generator data distributions. We also presented a simple framework that extends the proposed technique to commonly used deep generative models: GANs, VAEs, and Normalizing Flows. Experimental results indicate that gradient flows provide an excellent alternative methodology to refine generative models. Moving forward, we are considering several technical enhancements to improve DG$f$low's performance. At present, DG$f$low uses a stale estimate of the density-ratio, which could adversely affect sample evolution when the gradient flow is simulated for larger number of steps; how we can efficiently update this estimate is an open question. Another related question is when the evolution of the samples should be stopped; running chains for too long may modify characteristics of the original sample (e.g., orientation and color) which may be undesirable. This issue does not just affect DG$f$low; a method for automatically stopping sample evolution could improve results across refinement techniques.

## ACKNOWLEDGEMENTS

This research is supported by the National Research Foundation Singapore under its AI Singapore Programme (Award Number: AISG-RP-2019-011) to H. Soh. Thank you to J. Scarlett for his comments regarding the proofs.

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

## A    PROOFS

### A.1    LEMMA 3.1

*Proof.* Gradient flows in the Wasserstein space are of the form of the continuity equation (see Ambrosio et al. (2008), page 281), i.e,

$$\partial_t \rho_t + \nabla \cdot (\rho_t \mathbf{v}) = 0. \tag{15}$$

The velocity field $\mathbf{v}$ in Eq. (15) is given by

$$\mathbf{v}(\mathbf{x}) = -\nabla_{\mathbf{x}} \frac{\delta \mathcal{F}}{\delta \rho}(\rho), \tag{16}$$

where $\frac{\delta \mathcal{F}}{\delta \rho}(\rho)$ denotes the first variation of the functional $\mathcal{F}$. The first variation is defined as

$$\frac{d}{d\varepsilon} \mathcal{F}(\rho + \varepsilon \chi) \Big|_{\varepsilon=0} = \int \frac{\delta \mathcal{F}}{\delta \rho}(\rho) \chi, \tag{17}$$

where $\chi = \nu - \rho$ for some $\nu \in \mathcal{P}_2(\Omega)$.

Let's derive an expression for the the first variation of $\mathcal{F}$. In the following, we drop the notation for dependence on $\mathbf{x}$ for clarity,

$$\frac{d}{d\varepsilon} \mathcal{F}(\rho + \varepsilon \chi) \Big|_{\varepsilon=0} = \frac{d}{d\varepsilon} \int f\left(\frac{\rho + \varepsilon \chi}{\mu}\right) \mu + \gamma \int (\rho + \varepsilon \chi) \log(\rho + \varepsilon \chi) \Big|_{\varepsilon=0} \tag{18}$$

$$= \int f'\left(\frac{\rho + \varepsilon \chi}{\mu}\right) \chi + \gamma \int (\log(\rho + \varepsilon \chi) + 1) \chi \Big|_{\varepsilon=0} \tag{19}$$

$$= \int \left[ f'\left(\frac{\rho}{\mu}\right) + \gamma \log(\rho) + \gamma \right] \chi. \tag{20}$$

Substituting $\frac{\delta \mathcal{F}}{\delta \rho}(\rho)$ in Eq. (16) we get,

$$\mathbf{v}(\mathbf{x}) = -\nabla_{\mathbf{x}} \left[ f'\left(\frac{\rho}{\mu}\right) + \gamma \log(\rho) + \gamma \right] \tag{21}$$

$$= -\nabla_{\mathbf{x}} f'\left(\frac{\rho}{\mu}\right) - \frac{\gamma}{\rho} \nabla_{\mathbf{x}} \rho. \tag{22}$$

Substituting $\mathbf{v}$ in Eq. (15) we get the gradient flow,

$$\partial_t \rho_t - \nabla_{\mathbf{x}} \cdot \left( \rho_t \nabla_{\mathbf{x}} f'\left(\frac{\rho}{\mu}\right) + \rho_t \frac{\gamma}{\rho} \nabla_{\mathbf{x}} \rho \right) = 0 \tag{23}$$

$$\partial_t \rho_t(\mathbf{x}) - \nabla_{\mathbf{x}} \cdot \left( \rho_t \nabla_{\mathbf{x}} f'\left(\frac{\rho(\mathbf{x})}{\mu(\mathbf{x})}\right) \right) - \gamma \Delta_{\mathbf{xx}} \rho_t(\mathbf{x}) = 0, \tag{24}$$

where $\Delta_{\mathbf{xx}}$ and $\nabla_{\mathbf{x}} \cdot$ denote the Laplace and the divergence operators respectively.    $\square$

### A.2    LEMMA 3.2

*Proof.* Let $f$ be an integrable function on $\mathcal{X}$. If $\mathbf{J}_g$ has full column rank and $g$ is an injective function, then we have the following change-of-variables equation (Ben-Israel, 1999; Gemici et al., 2016),

$$\int_{\mathcal{X}} f(\mathbf{x}) d\mathbf{x} = \int_{\mathcal{Z}} (f \circ g)(\mathbf{z}) \sqrt{\det \mathbf{J}_g^\top \mathbf{J}_g(\mathbf{z})} d\mathbf{z}. \tag{25}$$

This implies that the infinitesimal volumes $d\mathbf{x}$ and $d\mathbf{z}$ are related as $d\mathbf{x} = \sqrt{\det \mathbf{J}_g^\top \mathbf{J}_g(\mathbf{z})}d\mathbf{z}$ and the densities $p_Z(\mathbf{z})$ and $q_X(\mathbf{x})$ are related as $p_Z(\mathbf{z}) = q_X(g(\mathbf{z}))\sqrt{\det \mathbf{J}_g^\top \mathbf{J}_g(\mathbf{z})}$. Similarly, $p_{\hat{Z}}(\hat{\mathbf{z}}) = q_{\hat{X}}(g(\hat{\mathbf{z}}))\sqrt{\det \mathbf{J}_g^\top \mathbf{J}_g(\hat{\mathbf{z}})}$. Finally, the density-ratio $p_{\hat{Z}}(\mathbf{u})/p_Z(\mathbf{u})$ at the point $\mathbf{u} \in \mathcal{Z}$ is given by

$$\frac{p_{\hat{Z}}(\mathbf{u})}{p_Z(\mathbf{u})} = \frac{q_{\hat{X}}(g(\mathbf{u}))\sqrt{\det \mathbf{J}_g^\top \mathbf{J}_g(\mathbf{u})}}{q_X(g(\mathbf{u}))\sqrt{\det \mathbf{J}_g^\top \mathbf{J}_g(\mathbf{u})}} = \frac{q_{\hat{X}}(g(\mathbf{u}))}{q_X(g(\mathbf{u}))}. \tag{26}$$

$\square$

## B  A DISCUSSION ON DG$f$LOW FOR WGAN

We apply DG$f$low to WGAN models by treating the output from their critics as the logit for the estimation of density-ratio. However, it is well-known that WGAN critics are not density-ratio estimators as they are trained to maximize the 1-Wasserstein distance with an unconstrained output. In this section, we provide theoretical justification for the good performance of DG$f$low on WGAN models. We show that DG$f$low is related to the gradient flow of the entropy-regularized 1-Wasserstein functional $\mathcal{F}_\mu^{\mathcal{W}} : \mathcal{P}_2(\Omega) \to \mathbb{R}$,

$$\mathcal{F}_\mu^{\mathcal{W}}(\rho) \triangleq \underbrace{\sup_{\|d\|_{\mathrm{Lip}} \leq 1} \int d(\mathbf{x})\,\mu(\mathbf{x})d\mathbf{x} - \int d(\mathbf{x})\,\rho(\mathbf{x})d\mathbf{x}}_{\text{1-Wasserstein distance}} + \gamma \underbrace{\int \rho(\mathbf{x}) \log \rho(\mathbf{x})d\mathbf{x}}_{\text{negative entropy}}, \tag{27}$$

where $\mu$ denotes the target density, $\|d\|_{\mathrm{Lip}}$ denotes the Lipschitz constant of the function $d$.

Let $d^*$ be the function that achieves the supremum in Eq. (27). This results in the functional,

$$\mathcal{F}_\mu^{\mathcal{W}}(\rho) = \int d^*(\mathbf{x})\,\mu(\mathbf{x})d\mathbf{x} - \int d^*(\mathbf{x})\,\rho(\mathbf{x})d\mathbf{x} + \gamma \int \rho(\mathbf{x}) \log \rho(\mathbf{x})d\mathbf{x}. \tag{28}$$

Following a similar derivation as in Appendix A.1, the gradient flow of $\mathcal{F}_\mu^{\mathcal{W}}(\rho)$ is given by the following PDE,

$$\partial_t \rho_t(\mathbf{x}) + \nabla_\mathbf{x} \cdot (\rho_t \nabla_\mathbf{x} d^*(\mathbf{x})) - \gamma \Delta_{\mathbf{xx}} \rho_t(\mathbf{x}) = 0. \tag{29}$$

If $d^*$ is approximated using the critic ($d_\phi$) of WGAN, we get the following gradient flow,

$$\partial_t \rho_t(\mathbf{x}) + \nabla_\mathbf{x} \cdot (\rho_t \nabla_\mathbf{x} d_\phi(\mathbf{x})) - \gamma \Delta_{\mathbf{xx}} \rho_t(\mathbf{x}) = 0, \tag{30}$$

which is same as the gradient flow of entropy-regularized $f$-divergence with $f = r \log r$ (i.e., the KL divergence) when $d_\phi$ is treated as a density-ratio estimator. The gradient flow of entropy-regularized $f$-divergence with $f = r \log r$ is simplified below,

$$\partial_t \rho_t(\mathbf{x}) - \nabla_\mathbf{x} \cdot (\rho_t \nabla_\mathbf{x} f'(\exp(-d_\phi(\mathbf{x})))) - \gamma \Delta_{\mathbf{xx}} \rho_t(\mathbf{x}) = 0 \tag{31}$$

$$\partial_t \rho_t(\mathbf{x}) - \nabla_\mathbf{x} \cdot (\rho_t \nabla_\mathbf{x} (\log(\exp(-d_\phi(\mathbf{x}))) + 1)) - \gamma \Delta_{\mathbf{xx}} \rho_t(\mathbf{x}) = 0 \tag{32}$$

$$\partial_t \rho_t(\mathbf{x}) + \nabla_\mathbf{x} \cdot (\rho_t \nabla_\mathbf{x} d_\phi(\mathbf{x})) - \gamma \Delta_{\mathbf{xx}} \rho_t(\mathbf{x}) = 0. \tag{33}$$

The equality of Eq. (30) and Eq. (33) implies that DG$f$low approximates the gradient flow of the 1-Wasserstein distance when the critic of WGAN is used for density-ratio estimation.

## C  FURTHER DISCUSSION ON RELATED WORK

**Energy-based & Score-based Generative Models**  DG$f$low is related to recently proposed energy-based generative models (Arbel et al., 2021; Deng et al., 2020) — one can view the energy functions used in these methods as a component that improves some base model. For example, the Generalized Energy-Based Model (GEBM) (Arbel et al., 2021) jointly trains an implicit generative model with an energy function and uses Langevin dynamics to sample from the combination

of the two. Similarly, in Deng et al. (2020), a discriminator that estimates the energy function is combined with a language model to train an energy-based text-generation model.

Score-based generative modeling (SBGM) (Song & Ermon, 2019; 2020) is another active area of research closely-related to energy-based models. Noise Conditional Score Network (NCSN) (Song & Ermon, 2019; 2020), a SBGM, trains a neural network to estimate the score function of a probability density at various noise levels. Once trained, this score network is used to evolve samples from noise to the data distribution using Langevin dynamics. NCSN can be viewed as a gradient flow that refines a sample right from noise to data; however, unlike DG$f$low, NCSN is a complete generative models in itself and not a sample refinement technique that can be applied to other generative models.

**Other Related Work**  Monte Carlo techniques have been used for improving various components in generative models, e.g., Grover et al. (2018) proposed Variational Rejection Sampling which performs rejection sampling in the latent space of VAEs to improve the variational posterior and Grover et al. (2019) used likelihood-free importance sampling for bias correction in generative models . Wu et al. (2020) proposed Latent Optimization GAN (LOGAN) which optimizes the latent vector as part of the training process unlike DG$f$low that refines the latent vector post training.

# D  IMPLEMENTATION DETAILS

## D.1  2D DATASETS

**Datasets**  The 25 Gaussians dataset was constructed by generating 100000 samples from a mixture of 25 equally likely 2D isotropic Gaussians with means $\{-4, -2, 0, 2, 4\} \times \{-4, -2, 0, 2, 4\} \subset \mathbb{R}^2$ and standard deviation 0.05. Once generated, the data-points were normalized by $2\sqrt{2}$ following Tanaka (2019). The 2DSwissroll dataset was constructed by first generating 100000 samples of the 3D swissroll dataset using `make_swiss_roll` from `scikit-learn` with `noise=0.25` and then only keeping dimensions $\{0, 2\}$. The generated samples were normalized by 7.5.

**Base Models**  We trained a WGAN-GP model for both the datasets. The generator was a fully-connected network with ReLU non-linearities that mapped $z \sim \mathcal{N}(0, I_{2 \times 2})$ to $x \in \mathbb{R}^2$. Similarly, the discriminator was a fully-connected network with ReLU non-linearities that mapped $x \in \mathbb{R}^2$ to $\mathbb{R}$. We refer the reader to Gulrajani et al. (2017) for the exact network structures. The gradient penalty factor was set to 10. The models were trained for 10K generator iterations with a batch size of 256 using the Adam optimizer with a learning rate of $10^{-4}$, $\beta_1 = 0.5$, and $\beta_1 = 0.9$. We updated the discriminator 5 times for each generator iteration.

**Hyperparameters**  We ran DOT for 100 steps and performed gradient descent using the Adam optimizer with a learning rate of 0.01 and $\beta = (0., 0.9)$ as suggested by Tanaka (2019). DDLS was run for 50 iterations with a step-size of 0.01 and the Gaussian noise was scaled by a factor of 0.1 as suggested by Che et al. (2020). For DG$f$low, we set the step-size $\eta = 0.01$, the number of steps $n = 100$, and the noise regularizer $\gamma = 0.01$. We used the output from the WGAN-GP discriminator directly as a logit for estimating the density ratio for DDLS and DG$f$low.

**Metrics**  We compared the different methods quantitatively on two metrics: % high quality samples and kernel density estimate (KDE) score. A sample is classified as a high quality sample if it lies within 4 standard deviations of its nearest Gaussian. The KDE score is computed by first estimating the KDE using generated samples and then computing the log-likelihood of the training samples under the KDE estimate. KDE was performed using `sklearn.neighbors.KernelDensity` with a Gaussian kernel and a kernel bandwidth of 0.1. The quantitative metrics were averaged over 10 runs with 5000 samples from each method.

## D.2  IMAGE EXPERIMENTS

**Datasets**  CIFAR10 (Krizhevsky et al., 2009) is a dataset of 60K natural RGB images of size $32 \times 32$ from 10 classes. STL10 is a dataset of 100K natural RGB images of size $96 \times 96$ from 10

Table 6: Network architectures used for MMDGAN and VAE models.

(a) Generator or Decoder

| Input Shape: (b, d, 1, 1) |
| --- |
| Upconv(256) |
| BatchNorm |
| ReLU |
| Upconv(128) |
| BatchNorm |
| ReLU |
| Upconv(64) |
| BatchNorm |
| ReLU |
| Upconv(3) |
| Tanh |
| Output Shape: (b, 3, 32, 32) |

(b) Discriminator or Encoder

| Input Shape: (b, 3, 32, 32) |
| --- |
| Conv(64) |
| LeakyReLU(0.2) |
| Conv(128) |
| BatchNorm |
| LeakyReLU(0.2) |
| Conv(256) |
| BatchNorm |
| LeakyReLU(0.2) |
| Conv(m) |
| Output Shape: (b, m, 1, 1) |

classes. We resized the STL10 (Coates et al., 2011) dataset to $48 \times 48$ for SNGAN and WGAN-GP, and to $32 \times 32$ for MMDGAN, OCFGAN-GP, and VAE since the respective base models were trained on these sizes.

**Base Models for CIFAR10**   We used the publicly available pre-trained models for WGAN-GP, SN-DCGAN (hi), and SN-DCGAN (ns). We refer the reader to Tanaka (2019) for exact details about these models. For SN-ResNet-GAN and OCFGAN-GP we used the pre-trained models from Miyato et al. (2018) and Ansari et al. (2020) respectively. We used the respective discriminators of SN-DCGAN (ns), SN-DCGAN (hi), and WGAN-GP for density-ratio estimation when refining their generators. For the SN-ResNet-GAN (hi) generator, we used SN-DCGAN (ns) discriminator as the non-saturating loss provides a better density-ratio estimation than a discriminator trained using the hinge loss.

We trained our own models for MMDGAN, VAE, and Glow. We used the generator and discriminator architectures shown in Table 6 for MMDGAN with $d = 32$. VAE used the same architecture with $d = 64$. Our Glow model was trained using the code available at `https://github.com/y0ast/Glow-PyTorch` with a batch size of 56 for 150 epochs. The density ratio correctors, $D_\lambda$ (see section 3.2), were initialized with the weights from the SN-DCGAN (ns) released by Tanaka (2019). $D_\lambda$ was then fine-tuned on images from SN-DCGAN (ns)'s generator and the generator being improved (e.g., MMDGAN and OCFGAN-GP) using SGD with a learning rate of $10^{-4}$ and momentum of 0.9. We fine-tuned $D_\lambda$ for 10000 iterations with a batch size of 64.

**Base Models for STL10**   We used the publicly available pre-trained models (Tanaka, 2019; Ansari et al., 2020) for WGAN-GP, SN-DCGAN (hi), SN-DCGAN (ns), and OCFGAN-GP. We trained our own models for MMDGAN and VAE with the same architecture and training details as CIFAR10. We fine-tuned the density ratio correctors for STL10 for 5000 iterations with other details being the same as CIFAR10.

**Hyperparameters**   We performed 25 updates of DG$f$low for CIFAR10 and STL10 with a step size of 0.1 for models that do not require density ratio corrections. For STL10 models that require a density ratio correction, we performed 15 updates with a step size of 0.05. The noise regularizer ($\gamma$), whenever used, was set to 0.01.

**Metrics**   We used the Fréchet Inception Distance (FID) (Heusel et al., 2017) and Inception Score (IS) (Salimans et al., 2016) metrics to evaluate the quality of generated samples before and after refinement. The IS denotes the confidence in classification of the generated samples using a pretrained InceptionV3 network whereas the FID is the Fréchet distance between multivariate Gaussians fitted to the 2048 dimensional feature vectors extracted from the InceptionV3 network for real and generated data. Both the metrics were computed using 50K samples for all the models, except Glow

Table 7: $f$-divergences and their derivatives.

| $f$-divergence | $f$ | $f'$ | $f''$ |
|---|---|---|---|
| KL | $r \log r$ | $\log r + 1$ | $\frac{1}{r}$ |
| JS | $r \log r - (r+1) \log \frac{r+1}{2}$ | $\log \frac{2r}{r+1}$ | $\frac{1}{r^2+r}$ |
| log D | $(r+1) \log(r+1) - 2 \log 2$ | $\log(r+1) + 1$ | $\frac{1}{r+1}$ |

where we used 10K samples. Following Tanaka (2019), we used the entire training and test set (60K images) for CIFAR10 and the entire unlabeled set (100K images) for STL10 as the set of real images used to compute FID.

### D.3 CHARACTER LEVEL LANGUAGE MODELING

**Dataset** We used the Billion Words dataset (Chelba et al., 2013) which was pre-processed into 32-character long strings.

**Base Model** Our generator was a 1D CNN which followed the architecture used by Gulrajani et al. (2017).

**Hyperparameters** We performed 50 updates of DG$f$low with a step size of 0.1 and noise factor $\gamma = 0$.

**Metrics** The JS-4 and JS-6 scores were computed using the code provided by Gulrajani et al. (2017) at `https://github.com/igul222/improved_wgan_training`. We used 10000 samples from the models to compute the JS-4 score.

## E ADDITIONAL RESULTS

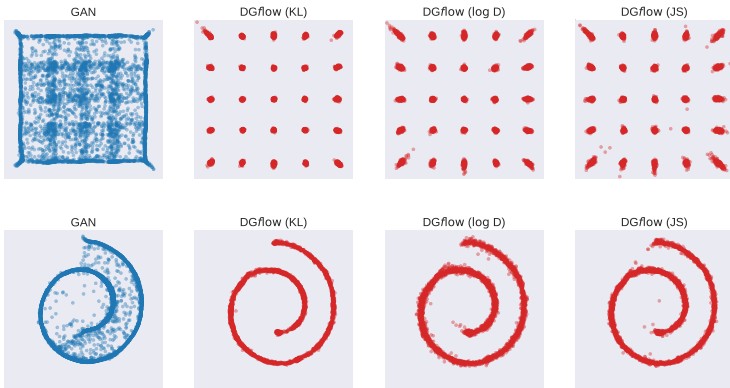

Figure 4: Qualitative comparison of DG$f$low with different $f$-divergences on the 25Gaussians and 2DSwissroll datasets.

Fig. 4 shows the samples generated by WGAN-GP (leftmost, blue) and refined samples generated using DG$f$low with different $f$-divergences (red). Fig. 5 shows the deterministic component, $-\nabla_{\mathbf{x}} f'(\rho_0/\mu)(\mathbf{x}_0)$, of the velocity for different $f$-divergences on the 2D datasets. Fig. 6 (right) shows the latent space distribution recovered by DG$f$low when applied in the latent space for the 2D datasets. This latent space is same as the one derived by Che et al. (2020), i.e., $p_t(\mathbf{z}) \propto p_Z(\mathbf{z}) \exp(d(g_\theta(\mathbf{z})))$ which is shown in Fig. 6 (left) for both datasets.

Table 11 shows the comparison of DG$f$low with DOT in terms of the inception score for the CIFAR10 and STL10 datasets. DG$f$low outperforms DOT significantly for all the base GAN models on both the datasets. Table 12 compares different variants of DG$f$low applied to MMDGAN, OCFGAN-GP, VAE, and Glow generators in terms of the inception score. DG$f$low leads to a

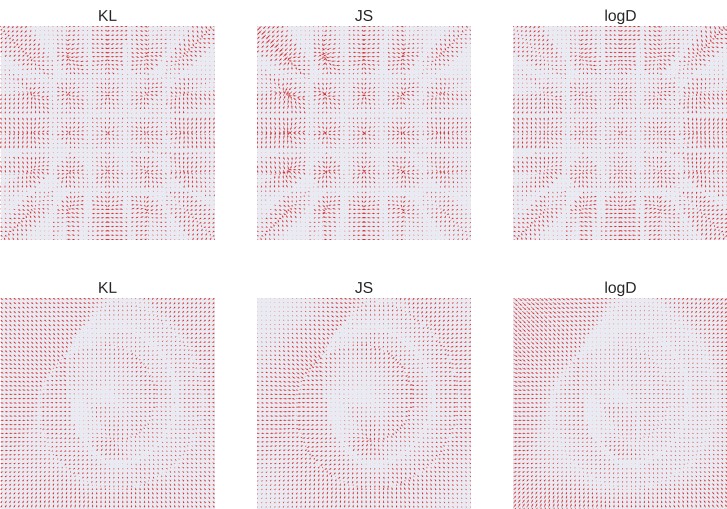

Figure 5: A vector plot showing the deterministic component of the velocity, i.e., the drift $-\nabla_{\mathbf{x}} f'(\rho_0/\mu)(\mathbf{x}_0)$, for different $f$-divergences on the 25Gaussians and 2DSwissroll dataset.

Table 8: Runtime comparison of DOT, DDLS, and DG$f$low$_{\text{(KL)}}$ on the 25Gaussians dataset. The runtime is averaged over 100 runs with standard deviation reported in parentheses.

| Method | Runtime (s) per 5K samples |
|---|---|
| DOT | 2.24 (0.18) |
| DDLS | 2.23 (0.14) |
| DG$f$low | 2.22 (0.15) |

significant improvement in the quality of samples for all the models. Tables 13 and 14 compare the deterministic variant of DG$f$low ($\gamma = 0$) against DOT and DDLS. These results show that the diffusion term only serves as an enhancement for DG$f$low, not a necessity, and it outperforms competing methods even without added noise. Table 15 shows the results of DG$f$low on MMDGAN, OCFGAN-GP, and VAE models when the SN-DCGAN (ns) discriminator is directly used as a density-ratio estimator without an additional density-ratio corrector. Figures 7, 8, 9, and 10 show the samples generated by the base model (left) and the refined samples (right) using DG$f$low for the CIFAR10 and STL10 datasets.

**Runtime** DG$f$low performs a backward pass through $d_\phi \circ g_\theta$ to compute the gradient of the density-ratio with respect to the latent vector. This results in the same runtime complexity as that of DOT and DDLS. Table 8 shows a comparison of the runtimes of DOT, DDLS, and, DG$f$low on the 25Gaussians dataset under same conditions. As expected, these refinement methods have similar runtimes in practice. The wall-clock time required for DG$f$low$_{\text{(KL)}}$ to refine 100 samples from different base models on the CIFAR10 and STL10 datasets is reported in tables 9 and 10.

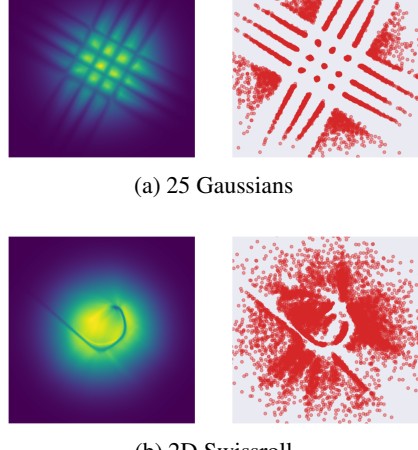

(a) 25 Gaussians

(b) 2D Swissroll

Figure 6: Latent space recovered by DG$f$low (right) for the 2D datasets is same as the one derived by Che et al. (2020) (left).

Table 9: Runtime of DG$f$low$_{(KL)}$ for models that do not require density-ratio correction on a single GeForce RTX 2080 Ti GPU. The runtime is averaged over 100 runs with standard deviation reported in parentheses.

| | Model | Runtime (s) per 100 samples |
|---|---|---|
| CIFAR10 | WGAN-GP | 0.897 (0.017) |
| | SN-DCGAN (hi) | 0.952 (0.008) |
| | SN-DCGAN (ns) | 0.952 (0.007) |
| | SN-ResNet-GAN | 1.982 (0.013) |
| STL10 | WGAN-GP | 1.376 (0.025) |
| | SN-DCGAN (hi) | 1.413 (0.015) |
| | SN-DCGAN (ns) | 1.415 (0.013) |

Table 10: Runtime of DG$f$low$_{(KL)}$ for models that require density-ratio correction on a single GeForce RTX 2080 Ti GPU. The runtime is averaged over 100 runs with standard deviation reported in parentheses.

| | Model | Runtime (s) per 100 samples |
|---|---|---|
| CIFAR10 | MMDGAN | 1.192 (0.007) |
| | OCFGAN-GP | 1.186 (0.011) |
| | VAE | 1.186 (0.012) |
| STL10 | MMDGAN | 1.036 (0.004) |
| | OCFGAN-GP | 1.029 (0.010) |
| | VAE | 1.028 (0.011) |

Table 11: Comparison of different variants of DG$f$low with DOT on the CIFAR10 and STL10 datasets. Higher scores are better.

| | Model | Inception Score | | | | |
|---|---|---|---|---|---|---|
| | | Base Model | DOT | DG$f$low$_{(KL)}$ | DG$f$low$_{(JS)}$ | DG$f$low$_{(\log D)}$ |
| CIFAR10 | WGAN-GP | 6.51 (.02) | 7.45 | **7.99 (.02)** | 7.71 (.02) | 7.11 (.03) |
| | SN-DCGAN (hi) | 7.35 (.03) | 8.02 | **8.13 (.02)** | 7.98 (.01) | 7.85 (.02) |
| | SN-DCGAN (ns) | 7.38 (.03) | 7.97 | **8.14 (.03)** | 7.98 (.04) | 7.94 (.01) |
| | SN-ResNet-GAN | 8.38 (.03) | – | **9.35 (.03)** | 9.13 (.04) | 9.05 (.03) |
| STL10 | WGAN-GP | 8.72 (.02) | 9.31 | **10.41 (.02)** | 8.85 (.06) | 9.80 (.03) |
| | SN-DCGAN (hi) | 8.77 (.03) | 9.35 | **9.74 (.04)** | 9.50 (.05) | 9.41 (.07) |
| | SN-DCGAN (ns) | 8.61 (.04) | 9.45 | **9.66 (.01)** | 9.49 (.03) | 9.18 (.03) |

Table 12: Comparison of different variants of DG$f$low applied to MMDGAN, OCFGAN-GP, VAE, and Glow models. Higher scores are better.

| | Model | Inception Score | | | |
|---|---|---|---|---|---|
| | | Base Model | DG$f$low$_{(KL)}$ | DG$f$low$_{(JS)}$ | DG$f$low$_{(\log D)}$ |
| CIFAR10 | MMDGAN | 5.74 (.02) | **6.27 (.05)** | 5.99 (.03) | 6.02 (.01) |
| | OCFGAN-GP | 6.52 (.02) | **7.21 (.05)** | 6.93 (.03) | 6.92 (.03) |
| | VAE | 3.20 (.01) | **3.85 (.01)** | 3.21 (.02) | 3.57 (.02) |
| | Glow | 3.64 (.02) | **4.57 (.02)** | 3.91 (.03) | 4.47 (.03) |
| CIFAR10 | MMDGAN | 6.07 (.02) | **6.16 (.01)** | 6.12 (.03) | 6.12 (.03) |
| | OCFGAN-GP | 7.09 (.01) | **7.46 (.04)** | 7.10 (.03) | 7.33 (.02) |
| | VAE | 3.25 (.01) | **3.72 (.04)** | 3.27 (.01) | 3.65 (.03) |

Table 13: Comparison of different variants of DG$f$low without diffusion (i.e., $\gamma = 0$) on the CIFAR10 and STL10 datasets. Lower scores are better.

| | Model | Fréchet Inception Distance | | | | |
|---|---|---|---|---|---|---|
| | | Base Model | DOT | DG$f$low$_{(KL)}$ | DG$f$low$_{(JS)}$ | DG$f$low$_{(\log D)}$ |
| CIFAR10 | WGAN-GP | 28.34 (.11) | 24.14 | 24.64 (.13) | **23.30 (.11)** | 24.42 (.19) |
| | SN-DCGAN (hi) | 20.67 (.09) | 17.12 | **15.79 (.07)** | 16.79 (.09) | 17.79 (.05) |
| | SN-DCGAN (ns) | 20.94 (.12) | 15.78 | **15.47 (.11)** | 16.32 (.11) | 16.97 (.08) |
| STL10 | WGAN-GP | 51.34 (.21) | 44.45 | **38.96 (.08)** | 50.44 (.09) | 39.35 (.12) |
| | SN-DCGAN (hi) | 40.82 (.16) | **34.85** | 35.18 (.09) | 36.53 (.13) | 36.75 (.13) |
| | SN-DCGAN (ns) | 41.83 (.20) | **34.84** | **34.81 (.08)** | 35.75 (.10) | 37.68 (.08) |

Table 14: Comparison of DDLS with DG$f$low (with and without diffusion) on the CIFAR10 dataset. Higher scores are better.

| Model | Inception Score |
|---|---|
| SN-ResNet-GAN (Miyato et al., 2018) | 8.22 (.05) |
| SN-ResNet-GAN + DDLS (cal) (Che et al., 2020) | 9.09 (.10) |
| SN-ResNet-GAN (our evaluation) | 8.38 (.03) |
| SN-ResNet-GAN + DG$f$low$_{(KL)}$ ($\gamma = 0$) | **9.35 (.04)** |
| SN-ResNet-GAN + DG$f$low$_{(KL)}$ ($\gamma = 0.01$) | **9.35 (.03)** |
| BigGAN | 9.22 |

Table 15: Comparison of different variants of DG$f$low applied to MMDGAN, OCFGAN-GP, and VAE models without density-ratio correction. Lower scores are better.

| | Model | Fréchet Inception Distance | | | |
|---|---|---|---|---|---|
| | | Base Model | KL | JS | log D |
| CIFAR10 | MMDGAN | 42.03 (.06) | **39.06 (.08)** | 39.68 (.06) | 39.47 (.07) |
| | OCFGAN-GP | 31.95 (.07) | 27.92 (.08) | 29.25 (.06) | **28.82 (.10)** |
| | VAE | 129.49 (.19) | **127.50 (.15)** | 128.24 (.11) | 128.3 (.14) |
| | Glow | 100.7 (.14) | **93.47 (.09)** | 97.50 (.11) | 97.78 (.14) |
| STL10 | MMDGAN | 47.22 (.04) | **45.75 (.10)** | 45.96 (.07) | 46.26 (.13) |
| | OCFGAN-GP | 36.60 (.15) | **34.17 (.18)** | 34.42 (.04) | 34.99 (.07) |
| | VAE | **150.49 (.07)** | 151.76 (.01) | 152.03 (.05) | 151.88 (.11) |

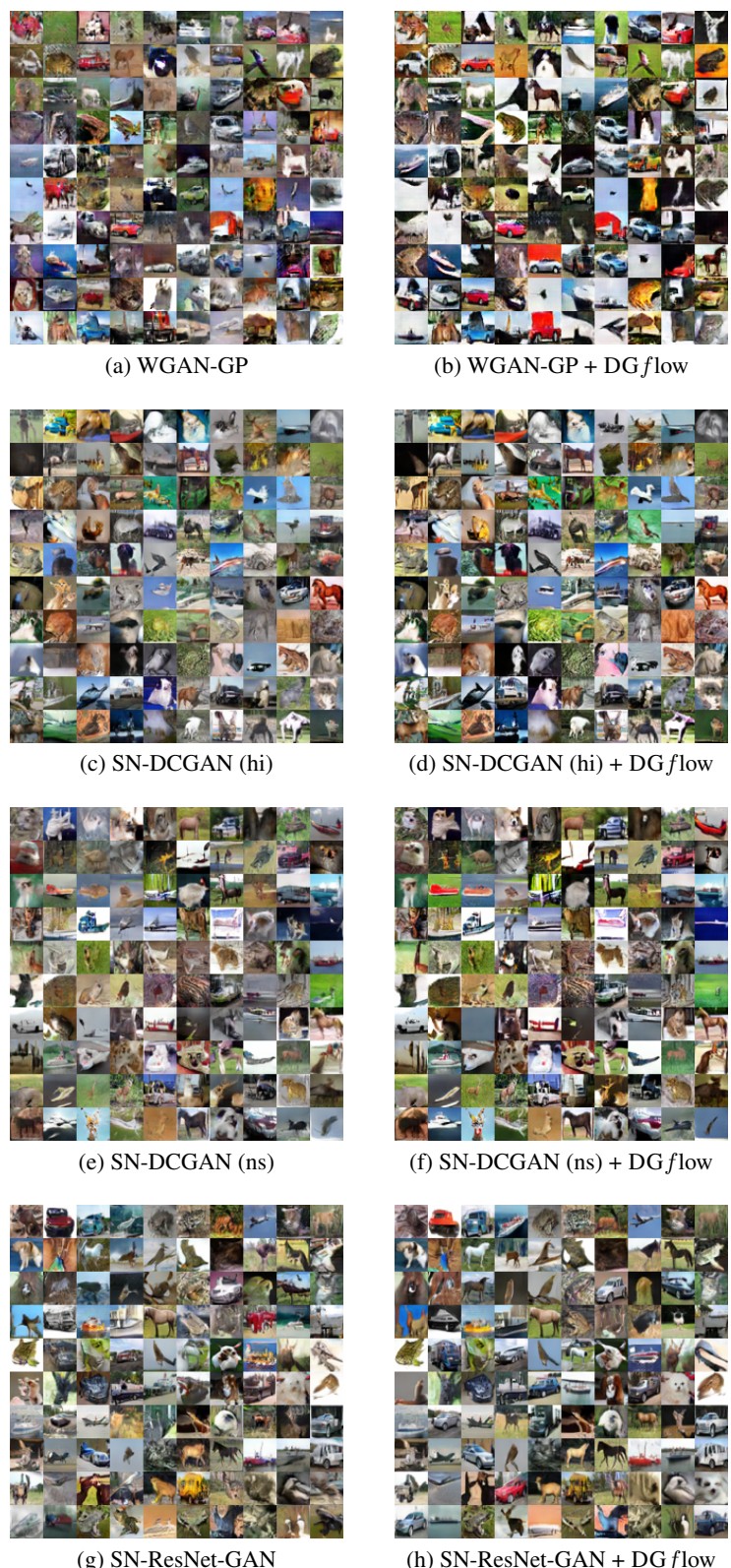

(a) WGAN-GP

(b) WGAN-GP + DG*f*low

(c) SN-DCGAN (hi)

(d) SN-DCGAN (hi) + DG*f*low

(e) SN-DCGAN (ns)

(f) SN-DCGAN (ns) + DG*f*low

(g) SN-ResNet-GAN

(h) SN-ResNet-GAN + DG*f*low

Figure 7: Samples from different models for the CIFAR10 dataset before (left) and after (right) refinement using DG*f*low.

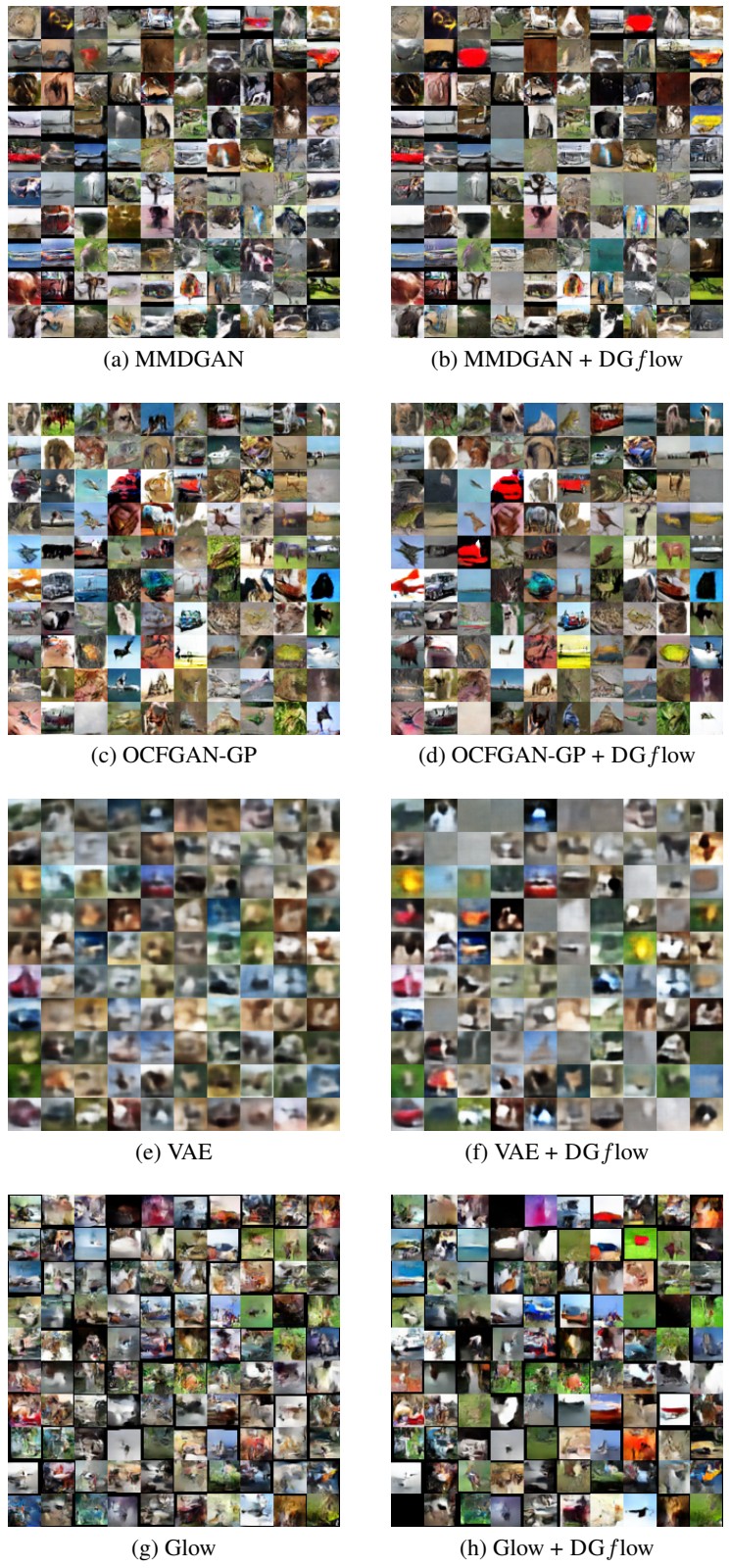

Figure 8: Samples from different models for the CIFAR10 dataset before (left) and after (right) refinement using DG*f*low.

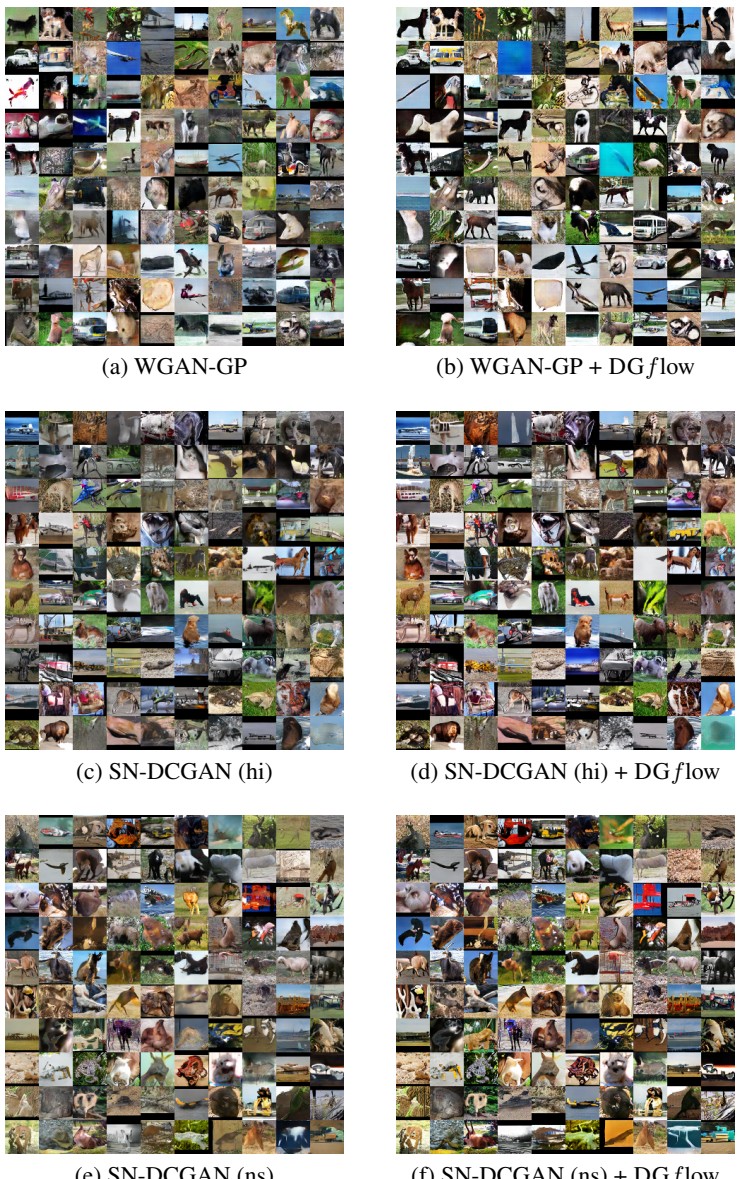

(a) WGAN-GP        (b) WGAN-GP + DG*f*low

(c) SN-DCGAN (hi)        (d) SN-DCGAN (hi) + DG*f*low

(e) SN-DCGAN (ns)        (f) SN-DCGAN (ns) + DG*f*low

Figure 9: Samples from different models for the STL10 dataset before (left) and after (right) refinement using DG*f*low.

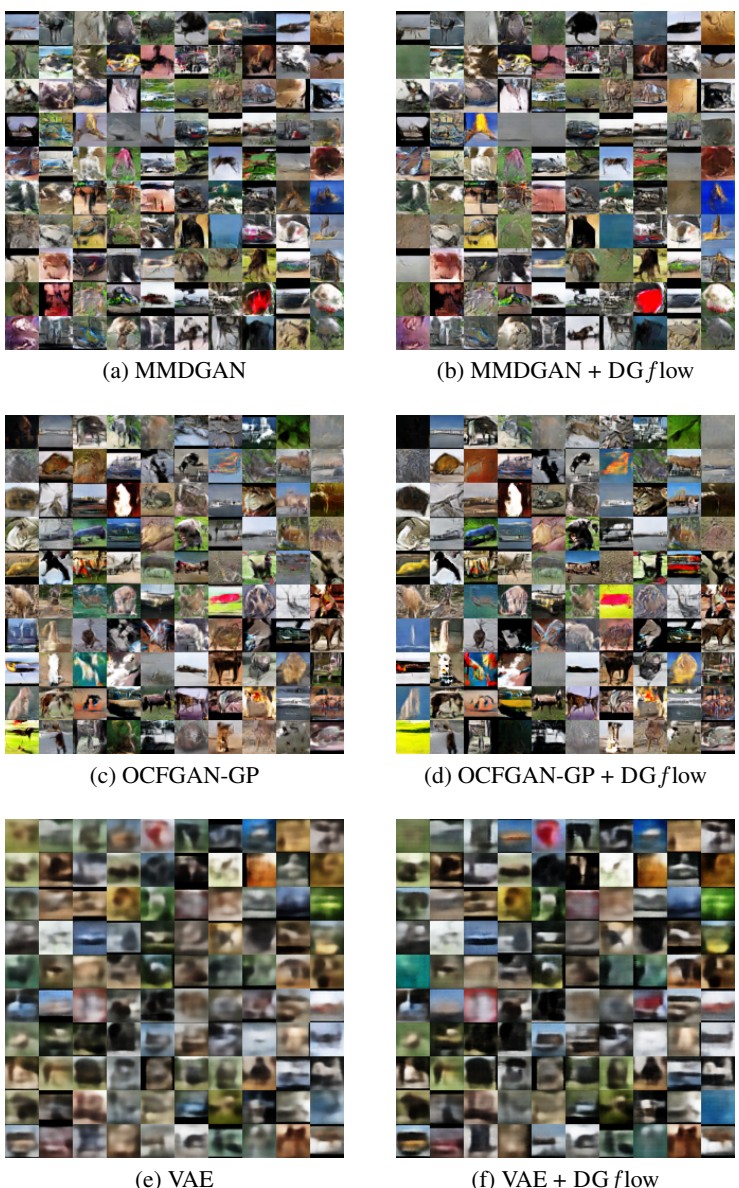

Figure 10: Samples from different models for the STL10 dataset before (left) and after (right) refinement using DG$f$low.

