# OpenReview forum: "Refining Deep Generative Models via Discriminator Gradient Flow"
_ICLR.cc/2021/Conference — ICLR 2021 Poster_

### Official Review · AnonReviewer4 · 2020-10-28
**Nice contribution in the field of sample refinement for Generative Modeling**

**Rating:** 7
**Confidence:** 2

**Review:**

Summary:
---------------
In this paper, the authors present a technique for refining the output of generative models by using gradients from discriminator to update the latent variables from which the generator produces a sample. The technique is more versatile than previous work since it can be applied to vector-valued discriminators, and does not restrict itself to scalar discriminators. The technique itself isn’t particularly complicated, yet delivers impressive results.

Pros:
-------
This work provides some sound theoretical foundations on which to build sample refinement, which isn’t specific to a single GAN architecture, but is applicable to a wide range of generative techniques. Although I’ll admit that I’m not familiar enough with stochastic differential equations to understand every aspect of the author’s theoretical justification, the idea of leveraging gradients from the discriminator to update not the generator (which lives in a high dimensional parameter space) but the latent variable (which lives in a much lower dimensional space) seems plausible.

In addition, one sees that the authors invested much thought and effort into the experiments section. I especially liked that they did language modeling, a task that shows a generative model works on domains other than just images. All told, after reading the experiments one is satisfied with the evidence of the effectiveness of the author’s technique.

Cons:
--------
The only major concern I have is that by only sampling images generated by updated latent variables, one loses diversity of generated images. A common experiment in some GAN papers was to choose two latent vectors $z_1$ and $z_2$, and show how the generator is able to smoothly interpolates between these two generated samples $g(z_1)$ and $g(z_2)$ as one generates samples with $g(\lambda z_1 + (1-\lambda) z_2), \lambda\in[0,1]$. It would be interesting to see if the latent refinement technique presented here also allows such smooth interpolation, or if as one starts with $\lambda z_1 + (1-\lambda) z_2$ and then refines, if the generated samples fall into a few “modes” of good samples while not generating the samples between the modes.

In 5.2 and 5.3, I would appreciate it if the authors could specify, as the authors did in 5.1, how many times the authors did the train a base model and then refine pipeline in order to obtain the means and standard deviations reported.

I would also appreciate it if the authors would report the JS-6 scores in the character level language modeling experiment. One gets the scores more or less for free, if runtime concerns are an issue, I can recommend this: <https://github.com/cseward/ngram_language_model> which does the same thing as the evaluation metric the authors used (<https://github.com/igul222/improved_wgan_training>) but is much faster by using C++ under the hood (there’s a python API) and saving the n-grams in a tree structure.

One last note: In our community, to get papers published one must generally demonstrate that a method works on some standard datasets, and authors are unfortunately yet understandably hesitant to discuss situations where a method fails to produce the desired results, as such a discussion could add arrows to the quiver of unduly critical reviewers. Such situations include datasets where the method doesn’t work, sensitivity to hyperparameters and  Yet it is exactly these counter-examples which often help further understanding of the method and pave the way for further advancements. Therefore I would humbly request that, if this paper is accepted, the authors also discuss any failure cases they may have found, as these would make the paper an even more compelling read.

Room for improvement:
---------------------------------
One of your claims is that “By refining inferior samples, our technique avoids expensive sample rejection used by previous methods.” Please consider reporting the runtime (i.e. wallclock time) of your method vs the methods you’re comparing against (such a report is fine in the appendix).

It’s a fact of life that most readers don’t want to delve into the mathematical details in order to understand a method, but like a few pretty pictures that give them at least an idea of what’s going on. I think the author’s paper would gain more traction in the community if in the introduction they could come up with some “eye candy” showing how the method allows the gradient to flow all the way from a discriminator to the latent variable, resulting in a better good generated sample.

---

> ### Author Response · Authors · 2020-11-17
> **Author response**
>
> We thank the reviewer for their positive remarks and constructive comments. Please see below for responses to specific issues:
>
> **Comment**: On the diversity of samples generated by latent interpolations.
> **Response**: We conducted an experiment on SN-ResNet-GAN trained on CIFAR10 as described by the reviewer. The images can be viewed [here](https://imgur.com/a/mmOskun) (anonymous imgur link). The first column shows the results of latent interpolation before refinement and each row shows the refinement of the sample in the first column. We note that the samples remain diverse and do not fall into modes of few good samples post-refinement. Instead, we see the refinement transforming less plausible samples into more plausible ones. That said, such latent interpolations may be less sensible for datasets studied in our paper (CIFAR10 & STL10) than for single object datasets such as CelebA (e.g., it isn't clear how to smoothly interpolate between a truck and a cat) . We conducted an additional preliminary experiment on SN-ResNet-GAN trained on the CelebA dataset. Similar to the previous experiment, the results ([imgur link](https://imgur.com/a/8GpocCY)) suggest that the samples remain relatively diverse.
>
> **Comment**: On the number of runs for the results reported in sections 5.2 and 5.3.
> **Response**: The scores for the image and text experiments (tables 2, 3, 4, and 5) have been averaged over 5 runs. We will clarify this in the revision.
>
> **Comment**: “I would also appreciate it if the authors would report the JS-6 scores in the character level language modeling experiment…”
> **Response**: Thank you for this suggestion. We have computed the JS-6 scores and will add the following to the paper.
>
> ```
> +---------------+----------------+
> |     Model     |      JS-6      |
> +---------------+----------------+
> | WGAN-GP       |  0.574 (.0015) |
> | DGflow (KL)   |  0.512 (.0012) |
> | DGflow (JS)   |  0.508 (.0011) |
> | DGflow (logD) |  0.506 (.0008) |
> +---------------+----------------+
> ```
>
> **Comment**: On the discussion of limitations and future research avenues.
> **Response**: Thank you for this suggestion. We can indeed add such samples (potentially into the appendix due to space). We will add a short discussion on limitations and potential future research avenues arising from our approach.
>
> **Comment**: "One of your claims is that 'By refining inferior samples, our technique avoids expensive sample rejection used by previous methods.' Please consider reporting the runtime...”
> **Response**: We will add a runtime comparison with DOT and DDLS baselines on the 2D datasets where we compared against them under the same conditions. We would like to clarify that both DOT and DDLS are refinement-based methods and *do not* perform sample rejections. One step of DOT, DDLS, and DGflow requires a backward pass through $d \circ g$ resulting in the same time complexity. The “previous methods” we are referring to in the above statement are DRS & MH-GAN that reject inferior samples instead of refining them which is wasteful. We will update this line to specify these methods to avoid any confusion.
>
> **Comment**: “... I think the author’s paper would gain more traction in the community if in the introduction they could come up with some “eye candy” showing how the method allows the gradient to flow all the way from a discriminator to the latent variable, resulting in a better good generated sample.”
> **Response**: Thanks for the suggestion; we agree that readers can benefit from a good visual. We are currently exploring how best to visualize DGflow, which we will add to the revision.
>
> We thank the reviewer again for their time and suggestions to improve the paper. We hope that we have satisfactorily answered your questions.

---

> > ### Comment · AnonReviewer4 · 2020-11-24
> > **Thanks for the experiments**
> >
> > I thank the authors for the latent interpolations experiments, they greatly help understand what the image refinement is doing. I think the authors are correct when saying "it isn't clear how to smoothly interpolate between a truck and a cat." So your refinement method will then no longer generating such funny-yet-unrealistic truck-cat combinations, and will generate either trucks or cats.
> >
> > It does seem from your experiments on CelebA that there too, one jumps from one thing to another since the faces don't change as smoothly as in the non-refined case. So I'm not convinced the samples are still really, truly diverse. Maybe it would be a good thing to investigate in future work.

---

> > > ### Author Response · Authors · 2020-11-24
> > > **Smoothness of refined interpolation would be interesting future work**
> > >
> > > An investigation of the smoothness of refined samples post interpolation would be an interesting direction for future work! The smoothness post-interpolation would likely depend on the underlying space learned by the discriminator-generator pair and not just on the refinement technique which merely explores this space. Potentially, some form of joint learning is required to generate smooth and refined interpolation.

---

### Official Review · AnonReviewer3 · 2020-10-28
**Refined generative models with Wasserstein gradient flows**

**Rating:** 7
**Confidence:** 3

**Review:**

# Summary
The paper introduces Wasserstein gradient flows for refining the samples of generative models. To achieve this, the authors demonstrate the gradient flow of divergences between two distribution in the Wasserstein space can be expressed as a stochastic differential equation whose dynamic mainly depends on the gradient of the divergence computed through the ratio between the two distributions. The authors observe the ratio between the generated distribution and the target distribution is provided by the discriminator and so the dynamic can be simulated via the stochastic Euler scheme. Instead of refining the distribution in the data space the authors suggest and show how to do it in the latent space. In addition, they expand their approach for non-GAN generative models, this setting requires to train (or just finetune)  another discriminator. They experimentally show that the proposed method improves the quality of the generated samples for a large panel of experimental settings. In particular, they slightly outperform other refining algorithms.
# Major comments
## Pros
I really enjoyed reading the paper, even if I had to read it twice to understand everything! The paper is well written and pleasant to read. In addition, the idea presented is very nice and is new to me. The main hypothesis made (that are not always valid) are clearly expressed and discussed. The experimental section convinced me the method is good at doing the job it is made for!

## Cons
I do not have strong concerns regarding your work. However, I am not very familiar with refining techniques. It is why I give "only" a 7/10.
I have however the following remarks:
1) When applying your method for a model that did not use a classifier to be trained, do you really think retraining another discriminator between the two generative models is better than just training the discriminator? Or is it because you do not suppose that you have access to the dataset but only two the two generative models? This is not very clear to me why you would like to estimate the ratio indirectly.
2) I would have liked to see the gradient flow in the 2D space fig 1. It would make a beautiful word and directly explicit how the method is working in this simple case. I don't know if this is easy to draw due to the randomness of the SDE though.
3) Why don't you compare to DDLS in table 2? I think the results could be made more readable by merging tables 2 and 3 and keeping only what is important in these tables.
4) I think discussing the intuition behind using different divergence and their impact on the refinement good also be interesting.
5) I would also like to see something about the induce "refinement" time, I suppose this is not too heavy but it would be interesting to have the information about that somewhere.
6) Experimenting on the violation of 3.2 would be interesting as well.

# Minor comments
focussed -> focused
Weiner -> Wiener
of particle -> of the particles

---

> ### Author Response · Authors · 2020-11-15
> **Author response**
>
> Thank you to the reviewer for their positive and constructive comments! Please see below for responses to your specific questions:
>
> **Comment**: “... do you really think retraining another discriminator between the two generative models is better than just training the discriminator? … why you would like to estimate the ratio indirectly.”
> **Response**: This is an interesting question; it turns out that neural density ratio estimators (that are binary classifiers) do not correctly estimate the density ratio when the two distributions are “far apart" (e.g., in terms of KL divergence). During our preliminary experiments, directly training the discriminator failed to improve the sample quality. We hypothesized that this was due to incorrect density ratio estimation, which led us to experiment with a density ratio "corrector", i.e., a density ratio estimator between the two generative models.
> We found very recent independent work on neural density ratio estimation (Rhodes et al., 2020) that corroborates this idea and provides additional insight: density ratio estimation using classifiers does not work well when distributions are far apart because it is too easy for the classifier to distinguish between such distributions. Rhodes et al. (2020) propose a technique termed telescoping density ratio estimation where intermediate distributions are constructed and the density ratio between $p_0$ and $p_n$ is estimated by a telescoping product,
> $$
> \frac{p_0(x)}{p_n(x)} = \frac{p_0(x)}{p_1(x)}\frac{p_1(x)}{p_2(x)}\dots\frac{p_{n-1}(x)}{p_n(x)},
> $$
> where $p_1,\dots,p_{n-1}$ are artificially constructed intermediate distributions. This telescoping product is similar to our approach. We treat the distribution from one of the generators as an intermediate distribution and estimate the final density ratio via a product of two ratios. We will update section 3.2 with this discussion to better substantiate our approach.
>
> **Comment**: “I would have liked to see the gradient flow in the 2D space…”
> **Response**: Thank you for the suggestion and we agree! As noted by the reviewer, it is difficult to depict the flow due to the stochastic component. We plan to plot the deterministic component of the velocity field (i.e., the drift component in Eq. (7)) and will add this vector plot to the paper.
>
> **Comment**: “Why don't you compare to DDLS in table 2? I think the results could be made more readable by merging tables 2 and 3 and keeping only what is important in these tables.”
> **Response**: The baseline results in table 2 & 3 are reproduced from their respective papers. Che et al. (2020) (DDLS) did not report results for the models reported in table 2. The base generative models reported in tables 2 and 3 are not directly comparable with each other. Therefore, we decided to split these results into two separate tables.
>
> **Comment**: “I would also like to see something about the induce "refinement" time…”
> **Response**: We will add details about the induced refinement time to the appendix. Briefly, one refinement step requires a backward pass through $d \circ g$ for the computation of gradient of the density ratio. The induced refinement time scales with the number of such steps.
>
> **Comment**: “Experimenting on the violation of 3.2 would be interesting as well.”
> **Response**: Lemma 3.2 requires the function $g$ to be injective and to have a Jacobian matrix with full column rank for all $z \in \mathcal{Z}$. Although this holds true for normalizing flows, it may not necessarily be true for generators used in GANs and VAEs. As such, many of our experiments already violate some conditions of Lemma 3.2, but the latent density ratio estimate still performs well.
>
> **Comment**: “focussed -> focused Weiner -> Wiener of particle -> of the particles”
> **Response**: We will fix these typographic errors in the revision.
>
> We thank the reviewer again for their positive comments and suggestions. We hope that we have satisfactorily answered their questions.
>
> [Rhodes et al., 2020] Benjamin Rhodes, Kai Xu, and Michael U. Gutmann. Telescoping density-ratio estimation. In NeurIPS, 2020.
> [Che et al., 2020] Tong Che, Ruixiang Zhang, Jascha Sohl-Dickstein, Hugo Larochelle, Liam Paull, Yuan Cao, and Yoshua Bengio. Your gan is secretly an energy-based model and you should use discriminator driven latent sampling. arXiv preprint arXiv:2003.06060, 2020.

---

### Official Review · AnonReviewer1 · 2020-10-29
**A principled methodology to improve samples generated from deep generative models**

**Rating:** 7
**Confidence:** 4

**Review:**

In this paper, authors have proposed to use Wasserstein gradient flows to update samples from deep generative models (DGM) to closer to the empirical data distribution which is expected to produce better generated samples. The Wasserstein gradient flows are induced from the entropy-regularized f-divergence functional between data the empirical data distribution and the to-be-learned distribution. The experimental results have shown improvement in the qualitative and quantitative results on real-world datasets of images and texts.

The writing is clear in general however the first paragraph of section 3.1 seems vague. Authors state that "experiments show
that the stale estimate ...  provides a good enough approximation for the purpose of refining samples" but then later they state "refining directly in the data-space using the stale estimate poses problems". It seems not consistent. Moreover, in the experimental sections, there is no result which refines "directly in the data-space".

It is not clear how to combine Eq. (8) with (11) to obtain the update on Eq. (12).  It will be more convinced if authors provide rigorous proof.

In the experimental section:
   - In Table 2, there is no result of FID for the baseline DDLS. Similarly, there is no result of IS for the baseline DOT.
   - Is it possible to use DOT/DDLS to apply for GANs with vector-valued critics (and/or VAE, Glow, etc.) by using auxiliary GDMs as with DGflow? If yes, what are their results in Table 4?

For images, KL divergence usually provides better results while  JS divergence is better for texts? Do authors have any insights?

---

> ### Author Response · Authors · 2020-11-14
> **Author response**
>
> We thank the reviewer for their time and positive comments on our work. Please see below for responses to specific questions:
>
> **Comment**: On refinement in the data-space using the stale estimate. “...in the experimental sections, there is no result which refines directly in the data-space.”
> **Response**: Thanks for pointing this out. We agree that the text is unclear here and will update the text to improve clarity. Refining directly in the data-space using the stale estimate causes problems for *high-dimensional datasets* (e.g., images). The experiments on the 2D datasets (subsection 5.1) have been conducted *directly in the data-space*. For image experiments, direct refinement in the RGB space led to visible degradation in the quality of the samples; thus, we performed the refinement in the latent space.
>
> **Comment**: “... how to combine Eq. (8) with (11) to obtain the update on Eq. (12).”
> **Response**: Eq. (12) is the latent space version of Eq. (8) with two differences: the initial distribution and the density ratio. The initial distribution of the latent space $p(u_{\tau_0})$ is equal to the prior distribution $p_Z(z)$ and the density ratio is obtained via Eq. (11):
>
> $$
> u_{\tau_{n+1}} = u_{\tau_n} - \eta\nabla_{u_{\tau_n}} f’\underset{\text{Eq. (11)}}{\underbrace{\left(p_{u_{\tau_n}}/p_{\hat{Z}}\right)(u_{\tau_n})}} + \sqrt{2\gamma\eta}\xi_{\tau_n}
> $$
>
> **Comment**: “In Table 2, there is no result of FID for the baseline DDLS. Similarly, there is no result of IS for the baseline DOT.”
> **Response**: The results for the baselines are reproduced from their respective papers. Che et al. (2020) (DDLS) did not report results for the models reported in Table 2. Similarly, Tanaka (2019) (DOT) did not report results for SN-ResNet-GAN. The corresponding inception score results for Tables 2 & 4 can be found in the Appendix.
>
> **Comment**: “Is it possible to use DOT/DDLS to apply for GANs with vector-valued critics (and/or VAE, Glow, etc.) by using auxiliary GDMs as with DGflow? If yes, what are their results in Table 4?”
> **Response**: The technique presented in subsection 3.2 is general and could, in theory, be applied to refinement methods that operate using the density-ratio (e.g., DDLS). However, we did not verify this experimentally.
>
> **Comment**: “For images, KL divergence usually provides better results while JS divergence is better for texts? Do authors have any insights?”
> **Response**: We thank the reviewer for raising this point. For the text dataset, we hypothesize that the better performance achieved by the JS divergence (JSD) is due to the evaluation metric; the JS-4 metric is based on the JSD between the 4-gram probabilities of the data generated by the model and the real data. That said, we believe that larger scale experiments on diverse generative tasks and evaluation metrics are required to draw a firm conclusion that JSD is indeed a better f-divergence for language modeling. Such an analysis of different f-divergences and their behavior for refining different types of datasets would constitute interesting future work.
>
> We thank the reviewer again and hope that we have satisfactorily answered their questions.
>
> [Che et al., 2020] Tong Che, Ruixiang Zhang, Jascha Sohl-Dickstein, Hugo Larochelle, Liam Paull, Yuan Cao, and Yoshua Bengio. Your gan is secretly an energy-based model and you should use discriminator driven latent sampling. arXiv preprint arXiv:2003.06060, 2020.
> [Tanaka, 2019] Akinori Tanaka. Discriminator optimal transport. In NeurIPS, 2019.

---

### Official Review · AnonReviewer2 · 2020-10-30
**The work presents an elegant framework to refine samples generated from a generative model.**

**Rating:** 7
**Confidence:** 4

**Review:**

Pros:
* The proposed framework seems principled and practical. Propagating generated samples to follow the data distribution by simulating the gradient flow of the f-divergence to the data distribution is reasonable, and the required quantity for simulation, i.e. the density ratio between the sample and data distributions, is readily given by the discriminator in GAN training.
* The presentation follows a clear logic flow, and related works are clearly connected. Experiment shows promising results.

Cons:
* Some statements can be made more precise and rigorous, up to my knowledge.
  - "The discriminator is trained to maximize this distance": although the discriminator is involved in a minimax optimization problem, it is the distributions-dependent optimal discriminator that defines a distance between two distributions. The minimax objective may not be a distance between two distributions given an arbitrary discriminator.
  - Eq. (1). Up to my knowledge, on a metric space, there is no formal definition of __gradient__. Even the concept of tangent vector is not defined on a metric space. Formally, a tangent vector involves a differential structure, so the space is often required to be a manifold. To define gradient, the space is further required to be a Riemannian manifold. What can be defined on a metric space is __gradient flow__, as __curves__ holding the intuition to minimize a given function steepest, and there are several formal descriptions on this intuition, e.g. the minimizing movement scheme. But the curves cannot be described using tangent vectors $x'(t)$ and gradients, as presented in Eq. (1).
  - Eq. (3) is specific to the 2-Wasserstein space.
* On Lemma 3.2.
The result is based on the rule of change of variables, Eq. (25). But if $g$ is not required to be injective, the right hand side of Eq. (25) needs to be multiplicated by the number of $z$'s that make $x = g(z)$ [Federer, 1969, "Geometric Measure Theory", Thm. 3.2.5]. So the lemma may need to be adjusted accordingly.
* It is favorable to cite the specific theorem/statement from works by Villani (2008) and Ambrosio et al. (2008) since they are huge books.
* On the method.
Although asymptotically the method guarantees that the sample distribution will converge to the data distribution, for each sample it may not converge and may traverse non-stop in the support of the data distribution. In other words, there exists nontrivial (non-zero) dynamics that keeps the data distribution stationary/invariant. Some examples in Fig. (2) already show this behavior to some extent, where all the samples along evolution seem realistic and differ in e.g. color or orientation. So is there a method to determine when to stop the evolution? Also, is it a problem that the evolution may change some attributes of the original sample?

=== EDIT: post rebuttal ===

Thanks for the response and addressing the issues for a more serious research paper.

---

> ### Author Response · Authors · 2020-11-13
> **Author response**
>
> We thank the reviewer for their positive comments and the thoroughness of their review. We especially appreciate the constructive comments on improving the rigor and preciseness of certain statements.  We have adapted the text based on your comments; please see below for details:
>
> **Comment**: “... The minimax objective may not be a distance between two distributions given an arbitrary discriminator.”
> **Response**: The word “distance” has been used here in an intuitive sense. To be more precise, we have replaced it with the following statement:
> “The generator seeks to generate samples that are similar to the real data by minimizing some measure of discrepancy between the generated samples and real samples. The discriminator is trained to distinguish the generated samples from the real samples.”
>
> **Comment**: “... on a metric space, there is no formal definition of gradient …”
> **Response**: We agree that the text here is informal; we intended it to be a high-level introduction to the concept of a gradient flow by connecting it to familiar quantities that are defined for a Euclidean space. We have modified the text to be more precise; instead of introducing $\mathrm{grad}$ in Eq (1) as a “notion of gradient” in the metric space, the revised text introduces Eq. (1) as the gradient flow for a Euclidean space and mentions that the idea of steepest descent curves can be extended to abstract metric spaces using the minimizing movement scheme.
>
> **Comment**: “Eq. (3) is specific to the 2-Wasserstein space.”
> **Response**: We have modified the text to specify that the density $\rho$ is in the 2-Wasserstein space.
>
> **Comment**: “On Lemma 3.2. The result is based on the rule of change of variables, Eq. (25). But if g is not required to be injective, the right hand side of Eq. (25) needs to be multiplicated by the number of z's that make x=g(z).”
> **Response**: We thank the reviewer for this comment. The assumption of full column rank of the Jacobian $J_g$ $\forall z \in \mathcal{Z}$ only implies that $g$ is locally injective. We have updated the Lemma to explicitly state injectivity of $g$ as one of the assumptions.
>
> **Comment**: “It is favorable to cite the specific theorem/statement from works by Villani (2008) and Ambrosio et al. (2008) since they are huge books.”
> **Response**: We have updated the references with specific theorem numbers.
>
> **Comment**: “... So is there a method to determine when to stop the evolution? Also, is it a problem that the evolution may change some attributes of the original sample?”
> **Response**:  Thanks for raising this point: to our knowledge, no method for stopping the evolution currently exists. In our experiments, we set the number of updates to a fixed value. As noted by the reviewer, increasing the number of update steps can change the attributes of the original sample. This behavior is not unique to DGflow and is present in other refinement methods as well. A method for stopping the evolution could improve results across the refinement techniques and would be interesting future work. We will add discussion around these points in the concluding section of our paper.
>
> We hope that we have satisfactorily addressed the reviewer’s concerns regarding the preciseness of certain statements and thank them again for their time.

---

### Official Review · AnonReviewer5 · 2020-11-06

**Rating:** 6
**Confidence:** 3

**Review:**

Summary

This paper proposes using gradient flows to improve samples from generative models. In practice, it uses the density ratio estimator from a GAN’s discriminator, in simulating the particle dynamics. The resulting algorithm updates latent variables based on a drift term, which is the gradient of the energy that combines the f-divergence and a negative entropy, and a diffusion term which adds a Gaussian noise. This method can be extended to other generative models such as VAE by training a density ratio corrector, similar to a discriminator.

The idea of introducing gradient flows for refining samples is theoretically very interesting, and the authors conducted a wide range of experiments to support the idea. However, I have major concerns regarding the actual algorithm and the evaluation. I would consider increasing my score if these can be well addressed.

Pros

1. Gradient flow provides a solid theory for improving samples from generative models, a direction that has attracted increasing attention recently. The idea proposed in this paper can help provide a unifying view of several recently proposed methods.
2. The paper is well-written, and it provides a good coverage of background materials.
3. It provides a wide range of experiments, from toy examples to image and text generation.

Cons

1. It is unclear to me why the proposed DGflow can outperform DDLS, which as the authors acknowledged, is equivalent to the DDLS when using the KL-divergence. More specifically how are DDLS and DGflow (KL) different in the experiments?
2. In Table 2, the FID for baseline models seem very different from numbers reported elsewhere (e.g., for SN-DCGAN(ns), the FID from Miyato et al. is 29.3, but here says 20.9).  With this discrepancy, it is difficult to evaluate the improvement from the proposed method.

Other comments:

1. The criticism of DDLS, that p_Z(z) might be undefined, seems unfair. The problem with uniform prior, e.g., can be handled by projected gradient descent. In addition, is there a similar term from entropy regularisation?
2. Consider discussing the relationship with using gradient flows for training as in [1]
3. Consider citing [2], which achieves the state-of-the-art GAN scores from incorporating refining latent in training.

[1] Deep Generative Learning via Variational Gradient Flow. Gao et al. 2019
[2] LOGAN: Latent optimisation for generative adversarial networks. Wu et al. 2019

---

> ### Author Response · Authors · 2020-11-12
> **Response to main comments**
>
> Thank you to the reviewer for their time and insightful comments. We are pleased to note the positive remarks on using gradient flows for refining generative models and how it unifies recently proposed methods. We respond to specific comments and concerns raised below:
>
> **Comment**: *"... updates latent variables based on a drift term, which is the gradient of the energy that combines the f-divergence and a negative entropy, and a diffusion term which adds a Gaussian noise."*
> **Response**: To clarify, the diffusion term is not combined with the f-divergence and negative entropy. Rather, the diffusion term is an outcome of the negative entropy regularization, which leads to the Laplacian term in the PDE (Eq. 6) and ultimately to the diffusion term in the SDE (Eq. 7).
>
> **Comment**: *“... why the proposed DGflow can outperform DDLS...”*
> **Response**: The DDLS results in Table 3 are reproduced from Che et al. (2020). In the absence of [official code](https://github.com/sodabeta7/gan_as_ebm), we could not perform a more thorough comparison on other models/datasets. DGflow (KL)'s Fokker-Planck Equation (FPE) can be shown to be formally equivalent to the FPE for Langevin dynamics used in DDLS by setting $f=r \log r$ and $\gamma=0$ in the functional (Eq. 5) and separating the terms as follows
> $$
> \mathcal{F}_\mu^{{KL}}(\rho) = \int\rho(z)\log\rho(z)dz - \int\rho(z)\log\mu(z)dz
> $$
> yielding the FPE
> $$
>     \partial_t\rho_t(z) +\nabla\cdot(\rho(z)\nabla\log\mu(z)) - \Delta\rho(z) = 0.
> $$
> However, this formulation requires an estimate of $\mu(z)$ which is set to $p_Z(z)\exp(d(G(z)))$ in DDLS. This leads to a difference in implementation compared to DGflow which only requires $\exp(d(G(z)))$, i.e., the density ratio. Moreover, DDLS also adds Gaussian noise to the generator output (Corollary 1 in Che et al., 2020) which is not done in DGflow. We hypothesize that these implementation differences are the reason for the differences in the scores.
>
> **Comment**: *“... FID for baseline models seem very different from numbers reported elsewhere…”*
> **Response**: We used the implementation and evaluation setup of Tanaka (2019) for our experiments. Tanaka (2019) reports an FID score of 20.7 for SN-DCGAN (ns) which is very similar to ours (20.9). Moreover, FID scores can also differ depending on the number of samples, deep learning library, and inception weights used to compute the scores (see Lee and Town, 2020). Rather than the absolute FID values, the refinement methods should be compared using the relative improvement in the FID scores, which reflects the quality of the refinement. We have also provided source code to replicate our experiments and for comparing other methods in the future.
>
>
> [Che et al., 2020] Tong Che, Ruixiang Zhang, Jascha Sohl-Dickstein, Hugo Larochelle, Liam Paull, Yuan Cao, and Yoshua Bengio. Your gan is secretly an energy-based model and you should use discriminator driven latent sampling. arXiv:2003.06060, 2020.
> [Tanaka, 2019] Akinori Tanaka. Discriminator optimal transport. In NeurIPS, 2019.
> [Gao et al., 2019] Yuan Gao, Yuling Jiao, Yang Wang, Yao Wang, Can Yang, and Shunkang Zhang. Deep generative learning via variational gradient flow. arXiv:1901.08469, 2019.
> [Gao et al., 2020] Yuan Gao, Jian Huang, Yuling Jiao, and Jin Liu. Learning implicit generative models with theoretical guarantees. arXiv:2002.02862, 2020.
> [Wu et al., 2019] Yan Wu, Jeff Donahue, David Balduzzi, Karen Simonyan, and Timothy Lillicrap. LOGAN: Latent optimisation for generative adversarial networks. In ICLR, 2020.
> [Lee and Town, 2020] Kwot Sin Lee, and Christopher Town. "Mimicry: Towards the Reproducibility of GAN Research." arXiv:2005.02494 (2020).

---

> > ### Author Response · Authors · 2020-11-12
> > **Response to other comments**
> >
> > **Comment**: *“The criticism of DDLS, that $p_Z(z)$ might be undefined, seems unfair ... is there a similar term from entropy regularisation?”*
> > **Response**: We thank the reviewer for this comment and will qualify our criticism with this detail:
> > “DDLS requires estimation of the score function to perform the update which becomes undefined if $z$ escapes the support of $p_Z(z)$; handling such cases would require techniques such as projected gradient descent.”
> >
> > Regarding entropy regularization, as noted above, it leads to the additive diffusion term (Eq. 7) and *not* a term similar to $p_Z(z)$ in DDLS.
> >
> > **Comment**: *“...  relationship with using gradient flows for training (Gao et al. 2019) …”*
> > **Response**: Section 2 of the paper briefly mentions that gradient flow of f-divergences have been used to train implicit generative models (Gao et al. 2019; 2020). We will add a brief discussion on this application of gradient flows and highlight the similarities/differences to our work.
> >
> > **Comment**: *Citation for relevant work (Wu et. al, 2019).*
> > **Response**: We thank the reviewer for pointing us to LOGAN (Wu et. al, 2019), which is relevant to our work. We will add a citation for the same.
> >
> > Again, we thank the reviewer for their time and we hope that we have satisfactorily addressed the above concerns. If so, we hope that the reviewer will consider revising their scores.
> >
> > [Che et al., 2020] Tong Che, Ruixiang Zhang, Jascha Sohl-Dickstein, Hugo Larochelle, Liam Paull, Yuan Cao, and Yoshua Bengio. Your gan is secretly an energy-based model and you should use discriminator driven latent sampling. arXiv:2003.06060, 2020.
> > [Tanaka, 2019] Akinori Tanaka. Discriminator optimal transport. In NeurIPS, 2019.
> > [Gao et al., 2019] Yuan Gao, Yuling Jiao, Yang Wang, Yao Wang, Can Yang, and Shunkang Zhang. Deep generative learning via variational gradient flow. arXiv:1901.08469, 2019.
> > [Gao et al., 2020] Yuan Gao, Jian Huang, Yuling Jiao, and Jin Liu. Learning implicit generative models with theoretical guarantees. arXiv:2002.02862, 2020.
> > [Wu et al., 2019] Yan Wu, Jeff Donahue, David Balduzzi, Karen Simonyan, and Timothy Lillicrap. LOGAN: Latent optimisation for generative adversarial networks. In ICLR, 2020.
> > [Lee and Town, 2020] Kwot Sin Lee, and Christopher Town. "Mimicry: Towards the Reproducibility of GAN Research." arXiv:2005.02494 (2020).

---

> > > ### Comment · AnonReviewer5 · 2020-11-24
> > > **Thanks for the reply**
> > >
> > > I thank the authors for their reply and clarification.
> > >
> > > Re. FID and IS baselines:
> > >
> > > Thanks for pointing out that these numbers indeed matches those from [Tanaka, 2019], which uses more samples (50k) for the inception metrics compared with many earlier papers. Nevertheless, I think it would be helpful to mention this evaluation detail in the main text to avoid confusion from the seemingly discrepancy from other well-known papers e.g., [Miyato, 2017].
> > >
> > > Re. Implementation resulted in the improvement over DDLS
> > >
> > > I appreciate that different perspectives can result in different implementations. However, it is still unclear to me where this advantage lies in practice. E.g., although adding noise to the generator output could in the end hamper DDLS' performance, this is unclear without, for example, ablation studies. This remains a concern due to the confounding of the method and its implementation.
> > >
> > > Taking these into further consideration, I raise my score to 6.
> > >
> > > [Miyato 2017] Spectral Normalization for Generative Adversarial Networks, ICLR 2017

---

### Decision · Program_Chairs · 2021-01-07
**Final Decision**

**Decision:**

Accept (Poster)

**Comment:**

This work improves deep generative models by applying Langevin dynamics to sample in the latent space. The authors test their method under different configurations (different loss functions) and various generative models (VAE, flow, besides GAN). Experimental results demonstrate the benefits of the proposed method in different generative tasks.

I tend to accept this solid work. I just have two suggestions: 1) the authors should discuss the connections and the differences between the proposed method and the energy-based methods like (Arbel et al., 2020) in-depth; 2) it may be more suitable to replace "Wasserstein gradient flow" with "Discriminator gradient flow" in the title.